# Better constrained climate sensitivity when accounting for dataset dependency on pattern effect estimates

Angshuman Modak[1,2] and Thorsten Mauritsen[1]

[1]Department of Meteorology, Stockholm University, Stockholm, Sweden
[2]Interdisciplinary Program in Climate Studies, IIT Bombay, Mumbai, India

**Correspondence:** Angshuman Modak, (angshuman.modak@gmail.com)

**Abstract.** The best estimate of Equilibrium climate sensitivity (ECS) constrained based on the instrumental record of the historical warming becomes coherent with other lines evidence when the dependence of radiative feedbacks on the pattern of surface temperature change (pattern effect) is incorporated. Pattern effect strength is usually estimated with atmosphere-only model simulations forced with observed historical sea-surface temperature (SST) and sea-ice change, and constant pre-industrial forcing. However, recent studies indicate that pattern effect estimates depend on the choice of SST boundary condition dataset, due to differences in the measurement sources and the techniques used to merge and construct them. Here, we systematically explore this dataset dependency by applying seven different observed SST datasets to the MPI-ESM1.2-LR model covering 1871-2017. We find that the pattern effect ranges from $-0.01 \pm 0.09$ Wm$^{-2}$K$^{-1}$ to $0.42 \pm 0.10$ Wm$^{-2}$K$^{-1}$ (standard error), whereby the commonly used AMIPII dataset produces by far the largest estimate. When accounting for the generally weaker pattern effect in MPI-ESM1.2-LR compared to other models, as well as dataset dependency and intermodel spread, we obtain a combined pattern effect estimate of 0.37 Wm$^{-2}$K$^{-1}$ [-0.14 to 0.88 Wm$^{-2}$K$^{-1}$] (5-95$^{th}$ percentiles) and a resulting instrumental record ECS estimate of 3.2 K [1.8 to 11.0 K], which as a result of the weaker pattern effect is slightly lower and better constrained than in previous studies.

## 1 Introduction

World governments are putting in extensive efforts to achieve the targets set in the Paris Agreement (2015), where the long-term goals are to limit the increase in global mean temperature to well below 2°C and pursuing efforts to limit the warming to 1.5°C above pre-industrial levels. However, the latest Intergovernmental Panel on Climate Change (IPCC, 2021) reported that our planet has already warmed by more than 1°C relative to the pre-industrial levels and it is likely that we might miss the 1.5°C target. To know what is required to meet the Paris Agreement goal it is imperative to better quantify and understand the ultimate amount of warming in response to a given forcing. Equilibrium climate sensitivity (ECS), defined as the long term warming resulting from a doubling of CO$_2$ concentration over pre-industrial levels, is a metric of central importance in

the quest to constrain projections of future global warming (Grose et al., 2018; Sherwood et al., 2020; Forster et al., 2021). Transient climate response (TCR) is another metric that is often used to study climate change which is however closely related to ECS (Huusko et al., 2021). Here, we investigate ECS constrained based on the instrumental record of historical warming including corrections for the effects on the radiation balance caused by the pattern of surface temperature change. Relative to earlier studies, we will do so by also accounting for uncertainty and biases caused by the underlying sea surface temperature reconstructions used in estimates of the pattern effect.

## 2 How historical warming is used to constrain equilibrium climate sensitivity

A linear energy budget framework incorporating global mean parameters is widely used to understand the response of the climate system to external perturbations such as a change in atmospheric composition (e.g., Gregory et al., 2002; Otto et al., 2013). It can be framed as $N = F + \lambda T$, where $N$ is the planetary energy imbalance which is generally defined as the change in net downward radiative flux at the top-of-the-atmosphere (TOA), $F$ is the external radiative forcing, defined as the effective radiative forcing (ERF) (Hansen et al., 2005; Forster et al., 2021), $T$ is the change in surface temperature, all relative to an unperturbed equilibrium climate state where $N = F = 0$. $\lambda$ is the climate feedback parameter which mediates how rapidly the climate system would get rid of the energy imbalance due to the external perturbation. ECS is related to the feedback parameter as, ECS = $-F_{2\times}/\lambda$, where $F_{2\times}$ is the forcing due to a doubled $CO_2$ concentration, such that:

$$\text{ECS} \approx \frac{F_{2\times}\Delta T}{\Delta F - \Delta N},$$

where the change in temperature, forcing and energy imbalance is taken between two periods, e.g. 1850-1900 and 2006-2019 denoted by $\Delta F$, $\Delta T$ and $\Delta N$.

The early best estimates of ECS based on the instrumental temperature record (Forster, 2016) are usually found to be at the lower end of the commonly accepted ECS likely range and triggered a lowering of the lower bound between the fourth and fifth IPCC assessment reports. For instance, Otto et al. (2013) best estimate of the historical energy budget constrained ECS is 2 K, whereas Lewis and Curry (2015) found a best estimate of 1.64 K. To reconcile this discrepancy, among other things the community looked into the concept of pattern effect which is not taken into account in the traditional energy balance framework. The rationale is that if the pattern of temperature response in the historical period is different in ways that affect the radiation balance from what it would be in the future, then constraining ECS based on the instrumental record could be biased. In addition there has been upward revisions to the temperature record through in-filling (e.g. Clarke and Richardson, 2021), downward revisions to total forcing and small revisions to estimates of energy imbalance (Bellouin et al., 2020; Von Schuckmann et al., 2020; IPCC, 2021).

Indeed, it has been identified that inference of $\lambda$ based on the instrumental record does not only rely on the global mean temperature but also on its spatial structure (Armour, 2017; Andrews et al., 2018; Lewis and Mauritsen, 2021; Fueglistaler and Silvers, 2021). Different patterns of temperature response lead to different circulation and cloud response. This in turn impact the energy budget and hence could induce dissimilar $\lambda$ (Zhou et al., 2016; Mauritsen, 2016; Ceppi et al., 2017). The

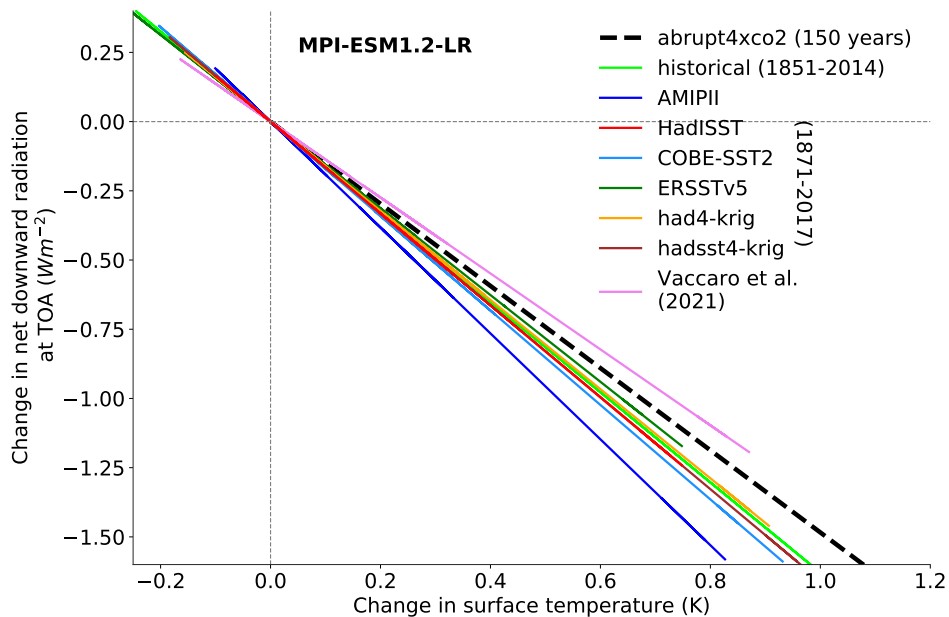

**Figure 1.** Linear fits from ordinary least square regression between the change in global-annual mean net downward radiation flux at TOA against the surface temperature change (Gregory et al., 2004) for different AOGCM (*abrupt4xCO2, historical*) and *observedSST-piForcing* Atmosphere-only GCM simulations (named as per datasets) shown in legend. The global-annual mean values are not shown but only the regression fits. The changes are relative to 1871-1900 mean. The y-intercept is adjusted such that all linear fits begins from origin. Pattern effect ($\Delta\lambda$) is the difference between the regression fits (solid lines) and the *abrupt4xCO2* fit (dashed line).

dependence of the radiative feedback on the spatial pattern of temperature response is coined "pattern effect" (Stevens et al., 2016).

Whereas a colder tropical eastern Pacific and Southern Ocean, and a stronger tropical western Pacific warming is observed over the historical period, Atmosphere-Ocean General Circulation Models (AOGCMs) simulate a long term climate response 50  (response to *abrupt4xCO2*) that resembles a temperature pattern similar to 'El-Nino Southern Oscillations' (ENSO) with relatively larger warming trends in the eastern Pacific and Southern Ocean (Andrews et al., 2015; Zhou et al., 2016; Dong et al., 2019; Sherwood et al., 2020; Watanabe et al., 2020). This difference in the distribution of temperature response induces a pattern effect since a warmer west Pacific and colder east Pacific leads to more stabilizing feedback while the simulated future temperature distribution would lead to a less stabilizing feedback (Zhou et al., 2016; Mauritsen, 2016; Ceppi et al., 2017; 55  Andrews and Webb, 2018; IPCC, 2021).

The strength of the pattern effect can be estimated as the difference in the radiative feedback obtained from a historical climate change simulation from that obtained from a long-term response simulation such as *abrupt4xCO2* (Figure 1, Armour, 2017; Andrews et al., 2018; Lewis and Mauritsen, 2021). One approach to determine an observationally constrained historical pattern effect is by prescribing the observed historical SST and sea-ice evolution such as the Atmospheric Model Intercom-

parison Project II (AMIPII) dataset to the AGCMs as a boundary condition with pre-industrial forcing (Andrews et al., 2018). This *observedSST-piForcing* configuration in principle simulates a TOA energy imbalance following the historical sea-surface temperature pattern evolution and hence facilitates the computation of historical $\lambda$ for the given model and the dataset (Figure 1 and Figure A1). Such estimates are model dependent, nevertheless most models yield a dampening pattern effect relative to the long-term *abrupt4xCO2* pattern based on the AMIPII dataset (e.g. Andrews et al., 2018).

However, this approach to estimate the pattern effect relies on the observed-reconstructed SST datasets applied to the AGCMs as boundary condition, and hence the estimates of pattern effect derived from such modeling experiments might depend on the applied SST dataset. Only a couple of studies (Lewis and Mauritsen, 2021; Fueglistaler and Silvers, 2021; Andrews et al., 2022) addressed the dataset dependency of the pattern effect in limited set ups. Lewis and Mauritsen (2021) compared only two, AMIPII and HadISST datasets, although they assessed the pattern effects derived using Green's function

approach with six other reconstructed datasets. They concluded that using alternative datasets yields a smaller pattern effect similar to that simulated by coupled climate models running the historical scenario. Fueglistaler and Silvers (2021), on the other hand, did not conduct simulations, but instead based on temperature metrics which rely on the region of deep convection and tropical average SSTs, a proxy for pattern effect (Dong et al., 2019), show that AMIPII stands out from the rest of the considered datasets.

In this study, we address this problem by employing seven datasets covering 1871-2017 to the MPI-ESM1.2-LR model. We simulate a range of pattern effect estimates based on the datasets. The resulting dataset dependent estimates of the pattern effect is then extended to include model dependency estimated by Andrews et al. (2022), to update the ECS estimates constrained by the instrumental record reported in Forster et al. (2021).

## 3   Model, datasets and experiments

The MPI-ESM1.2-LR climate model (Mauritsen et al., 2019) participated in the sixth phase of the coupled model intercomparison project (CMIP6). The atmosphere model component is ECHAM6.3. Through surface exchange of mass, momentum and heat, it is coupled to the land model JSBACH3.2. The horizontal resolution is spectral T63, corresponding to approximately 200 km grid spacing, while there are 47 hybrid-sigma pressure levels in the vertical. We configure MPI-ESM1.2-LR to run in atmosphere-only mode where the SST boundary conditions are prescribed based on the observed historical surface temperature

evolution.

   The seven different observed-reconstructed SST datasets applied as boundary conditions are HadISST1 (Rayner et al., 2003), AMIPII (Hurrell et al., 2008), COBE-SST2 (Hirahara et al., 2014), ERSSTv5 (Huang et al., 2017), had4krig and had4sst4krig (Cowtan and Way, 2014) and Vaccaro2021 (Vaccaro et al., 2021) (Table A1). All of these datasets are globally complete in-filled fields of SST in monthly resolution from 1871 to 2017. The had4krig, had4sst4krig and Vaccaro2021 datasets are

available only as anomalies relative to a base period. We have used the corresponding base period from COBE-SST2 to create the absolute temperatures. The differences among these datasets arise from the differences in the 1) measurement sources that they are constructed from, the bulk of which is common, 2) assimilation and bias correction methods applied, and 3) infilling

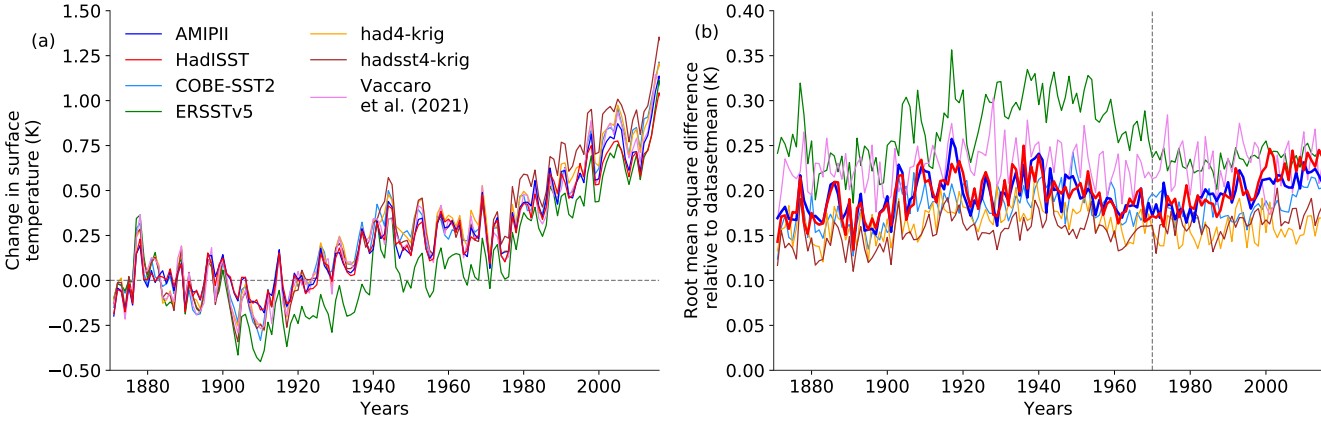

**Figure 2.** Evolution of simulated global mean surface temperature anomaly (a) and root-mean-squared-difference (RMSD) of the temperature anomaly for the seven *observedSST-piForcing* simulations relative to the dataset mean (b). The dashed gary line in (b) marks year-1970. The RMSD are calculated over each ocean grid with the land masked. The temperature anomalies are relative to 1871-1900 mean.

methods employed which aim to construct a spatially complete dataset. For instance, had4krig and had4sst4krig employs an optimal interpolation algorithm known as kriging (Cressie, 1990) to HadCRUT4 data whereas Vaccaro2021 employs a method based on Gaussian graphical models to the raw HadCRUT4.6 (Vaccaro et al., 2021). A comparison of the dataset properties is given in Table A1.

The datasets are regridded to the Gaussian grid corresponding to the T63 resolution of MPI-ESM1.2-LR. We performed a bi-linear interpolation on the datasets. The same is done for the AMIPII sea-ice data which is set as the boundary condition for sea-ice concentration in all the simulations in order to isolate the effect of the SSTs. We apply pre-industrial forcing to these set of simulations and named them as *observedSST-piForcing*. We perform an ensemble of five *observedSST-piForcing* simulations for each dataset. Unless otherwise specified, throughout the text, the displayed results are based on the mean of the 5 ensemble simulations while the uncertainty denotes the standard error from the ordinary least square (OLS) regression and the ranges are the 5-95$^{th}$ percentiles.

As expected, the historical evolution of the global annual mean surface temperature anomaly from the *observedSST-piForcing* simulations are similar (Figure 2a). The differences are due to differences in the way the SST fields are reconstructed and partly due to the land surface warming which evolves on its own in these simulations. The temperature anomaly in ERSSTv5 forced *observedSST-piForcing* simulation is lower compared to others, consistent with Fueglistaler and Silvers (2021). This is related to ship SST bias corrections made to the temperature during the 1880-1940s and 1950-1960s (Huang et al., 2017) (Figure 2a).

The root-mean-squared-difference of surface temperature anomaly between the individual *observedSST-piForcing* simulations and their mean diverges most in the case of ERSSTv5 until the 1970s (Figure 2b-d), which is probably mostly due to the lower global mean warming in that dataset. The differences reduces to within 0.15 K after 1970s. Although there is a close agreement in the global mean evolution of surface warming, we find apparent regional differences in the temperature anomaly

trends shown in Figure 3. These differences in the pattern of surface temperature change could lead to different estimates of pattern effect.

We obtain $\lambda_{4xCO2}$ and $\lambda_{hist}$ from the corresponding AOGCM simulations *piControl, abrupt4xCO2, historical* which are available from the CMIP6 database for MPI-ESM1.2-LR. We perform 150-year OLS regression of the global annual mean TOA net radiative flux change ($N$) against the surface temperature change ($T$) to estimate $\lambda_{4xCO2}$ as -1.48 $\pm$ 0.03 Wm$^{-2}$K$^{-1}$ (Gregory et al., 2004). The changes are between the first 150 years of *abrupt4xCO2* simulation and the mean of the last 500 years of *piControl* simulation. Note, since we use surface temperature in our analysis, our estimate of $\lambda_{4xCO2}$ for MPI-

ESM1.2-LR is larger than the estimate from Zelinka et al. (2020) and Andrews et al. (2022) which uses surface air temperature instead. For $\lambda_{hist}$, we take the changes between the global annual mean $N$ and $T$ of the entire period 1851-2014 of *historical* simulation relative to the mean of first 50 years of the same. Since, there are 10 ensemble members present in the *historical* simulation, we use the ensemble mean in our calculation. In addition, we conduct a simulation in fixed-SST configuration, but with evolving historical forcings to evaluate the effective radiative forcing ($F$) from 1851-2014. To calculate $F$, we account

for the land surface warming: we subtract the product of $\lambda_{4xCO2}$ and the land surface warming from $N$ simulated by historical simulation in fixed-SST configuration (Hansen et al., 2005; Modak et al., 2018). We then regress $N - F$ against $T$ to estimate $\lambda_{hist}$ as -1.64 $\pm$ 0.07 Wm$^{-2}$K$^{-1}$ (Figure 1).

## 4    Results and discussion

In the following we present the pattern effect estimated on the seven SST datasets, along with an inference of the unforced part.

Then we discuss the differences between the datasets, and close by evaluating the impact of these new findings on estimates of ECS from historical warming.

### 4.1    Total pattern effect

The difference in feedback between the *abrupt4xCO2* and the *observedSST-piForcing* AGCM simulations is defined as the total pattern effect (Figure 4a). We call it "total" to highlight the fact that the *observedSST-piForcing* AGCM simulations

encapsulates the radiative effect of spatial distribution of temperature which depends on the all external forcings during the historical period as well as internal variability (Gregory et al., 2019; Seager et al., 2019; Watanabe et al., 2020; Lewis and Mauritsen, 2021). On the other hand, for a given model the forced pattern effect can be derived from an ensemble of historical simulations. This pattern effect is the result of all forcing agents applied in the respective historical simulation. The difference between the "total" and "forced" could then be interpreted to be due to internal variability (discussed in next section). It is worth

mentioning that recent studies (e.g. Seager et al., 2019) while interpreting the temperature gradient in the equatorial Pacific proposed that rising greenhouse gases is the cause for the observed temeprature gradient. However, other studies supports natural forcing  internal variability are equally possible causes (e.g. Gregory et al., 2019; Watanabe et al., 2020; Olonscheck et al., 2020).

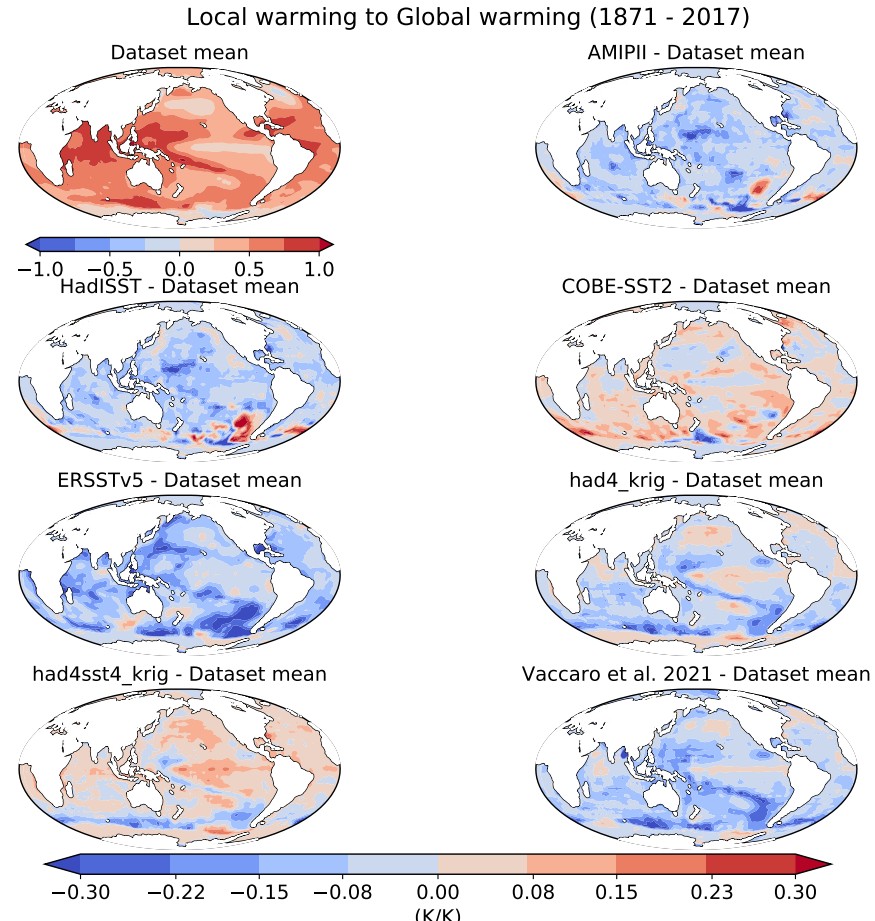

**Figure 3.** Change in the pattern of sea-surface temperature (change in T against change in global-mean T; in K/K) from 1871-2017 for each of the *observedSST-piForcing* simulations relative to the dataset mean.

Depending on the underlying dataset, the strength of the total pattern effect ranges from -0.01 $\pm$ 0.09 Wm$^{-2}$K$^{-1}$ to 0.42
$\pm$ 0.10 Wm$^{-2}$K$^{-1}$ over 1871-2017 while the mean estimate is 0.20 Wm$^{-2}$K$^{-1}$ [0.01 to 0.39 Wm$^{-2}$K$^{-1}$] (Figure 4a). The uncertainty in the pattern effect estimates are calculated by adding the errors from $\lambda_{4xCO2}$ and respective $\lambda$ from *observedSST-piForcing* simulations in quadrature. A positive pattern effect would lead to a relatively less stabilizing feedback i.e. a less negative $\lambda$ consequently a higher climate sensitivity. The mean estimate of the pattern effect derived from different dataset is less than the multi-model mean estimate of Andrews et al. (2022) and the mean estimate considered in IPCC AR6 (IPCC,
2021). Nevertheless, here one must keep in mind that MPI-ESM1.2-LR produces a slightly weaker pattern effect than on average by other models. For example, Andrews et al. (2022) reported a value of 0.56 Wm$^{-2}$K$^{-1}$ for ECHAM6.3, the atmosphere

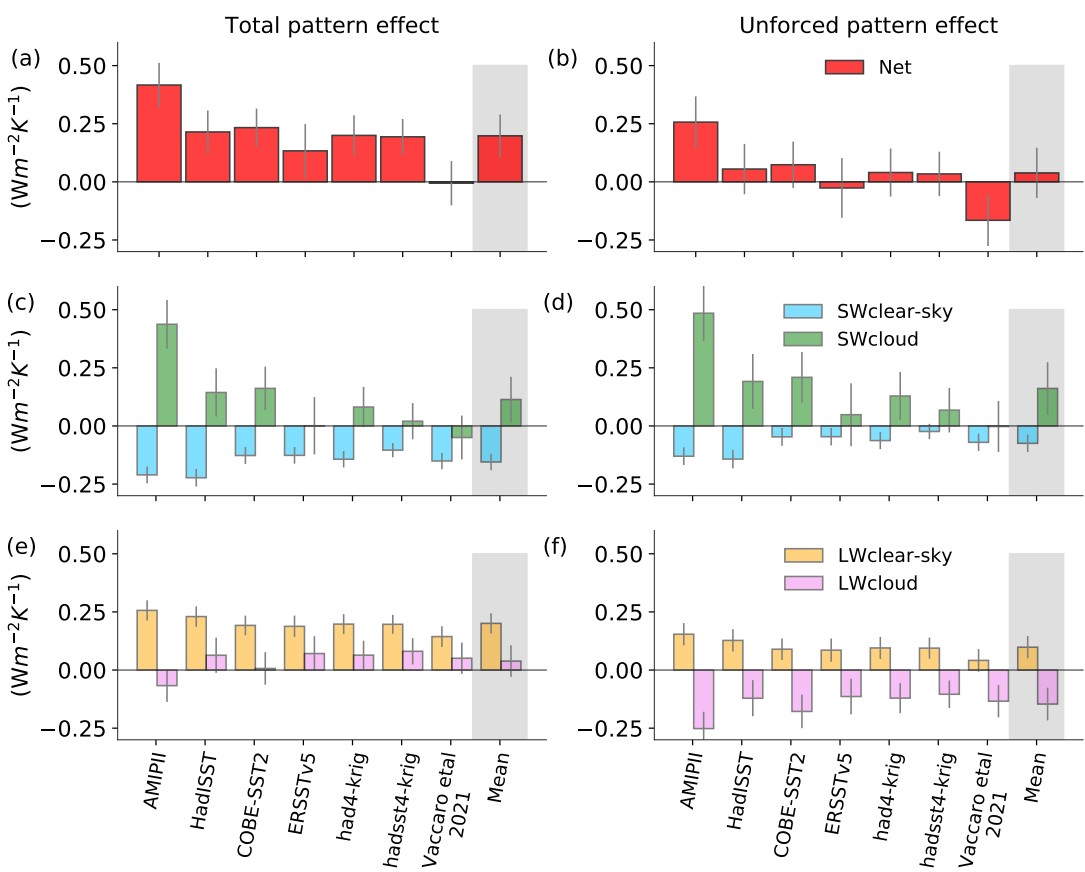

**Figure 4.** Estimates of total- (a,c,e) and unforced pattern effect (b,d,f) and its shortwave (SW) and longwave (LW) radiative components from the seven *observedSST-piForcing* simulations from 1871-2017. Also shown is the dataset mean estimate in gray shaded region. The error bars show $\pm 1$ standard error.

component of MPI-ESM1.2-LR, whereas the multi-model mean was larger at 0.70 $\mathrm{Wm^{-2}K^{-1}}$ with AMIPII (more discussion in section 4.5). The differences in pattern effect among the *observedSST-piForcing* simulations arises primarily due the differences in the SW-cloud feedback (Figure 4c). This is expected as the spatial pattern of temperature response primarily drives

the circulation and the cloud response (Zhou et al., 2016; Ceppi et al., 2017; Mauritsen, 2016; Andrews and Webb, 2018). The large positive SW-cloud pattern effect in case of AMIPII compared to the rest of the datasets is responsible for inducing the overall larger net pattern effect.

    We inferred the total pattern effect from Lewis and Mauritsen (2021) for the in-common SST datasets which they calculated based on CAM5.3 Green's function (Figure A2). We find that their estimates of total pattern effect are substantially different

from our estimates for some of the SST datasets. The differences could be either because of the Green's function that is applied in Lewis and Mauritsen (2021) is derived from a different model (CAM5.3 Green's function applied to ECHAM6.3) and

different models produce different pattern effects, or could be due to its inherent limitations (Zhou et al., 2017). However, we find that the uncertainty in the pattern effect estimates across the in-common SST datasets are of similar magnitudes between the studies. We plan to further address the comparison with the estimates derived from Green's function in a future study.

## 4.2 Unforced pattern effect

The unforced pattern effect which represents the pattern effect due to the internal variability in the climate system is defined as the difference in feedback between the *historical* and the *observedSST-piForcing* simulations, as discussed in the previous section. The rationale is that the *observedSST-piForcing* simulations could be one of the possible trajectory other than the full-coupled *historical* simulation.

The dataset mean estimate of the unforced pattern effect is quite small in agreement with Lewis and Mauritsen (2021) although it ranges from $-0.17 \pm 0.11 \, \mathrm{Wm^{-2}K^{-1}}$ to $0.26 \pm 0.11 \, \mathrm{Wm^{-2}K^{-1}}$: except AMIPII and Vaccaro2021 based estimates which lies at the two ends, all other datasets induce small values (Figure 4b). In the recent decades we find strong dampening of the unforced pattern effect. The datatset mean estimate for the period 1970-2017 is $0.48 \pm 0.19 \, \mathrm{Wm^{-2}K^{-1}}$, for 1980-2017 it is $0.59 \pm 0.25 \, \mathrm{Wm^{-2}K^{-1}}$ and for the most recent period 2000-2017 it is $0.29 \pm 0.48 \, \mathrm{Wm^{-2}K^{-1}}$. Thus, when inspecting short periods the unforced pattern effect can be substantial but across the century we find only small pattern effects.

## 4.3 Dataset differences

To investigate the differences in the strength of the pattern effect among the datasets, we compare the temperature trends over the Indo-Pacific Warm Pool (IPWP), Equatorial West Pacific (EWP), Equatorial East Pacific (EEP) and Southern ocean (SO). The temperature trends over these key regions in the historical period are typically in contrast to that in the long term response (e.g. Andrews et al., 2018; Dong et al., 2019; Lewis and Mauritsen, 2021; Fueglistaler and Silvers, 2021). We find that the local warming to the warming over 50°S-50°N from 1871-2017 in each of these regions have significant differences in some cases among the datasets (Figure 5). For instance, over the IPWP region, the ERSSTv5 dataset shows significant difference compared to the HadISST, COBE-SST2, had4krig, hadsst4krig or Vaccaro2021. The local to warming over 50°S-50°N is quite different for the Vaccaro21 dataset than the other datasets in all regions except EEP.

Past studies showed that the relative warming of the IPWP compared to the rest of the oceans or to the tropics, could influence the strength of the pattern effect (Dong et al., 2019; Fueglistaler and Silvers, 2021; Lewis and Mauritsen, 2021). Dong et al. (2019) advocated the IPWP to govern the strength of the pattern effect as they find a strong dependence of the TOA radiation balance on the temperature over IPWP. However, later it was found that such clear dependence is not supported by the CMIP6 models (Dong et al., 2020; Lewis and Mauritsen, 2021). Across the datasets, we find a comparative correlation between the pattern effect and the regional warming relative to the 50°S-50°N: 0.64 over IPWP, 0.31 over EWP, -0.43 over EEP and 0.69 over SO (Figure 6). Although AMIPII stands out from the rest of the datasets in its pattern effect strength, it does not show substantial difference in the relative warming except over the EEP and SO (Figure 6c). However, both over EEP and SO, even HadISST1 which simulates a relatively weak pattern effect has a relative warming similar to AMIPII. Thus, it is difficult to link

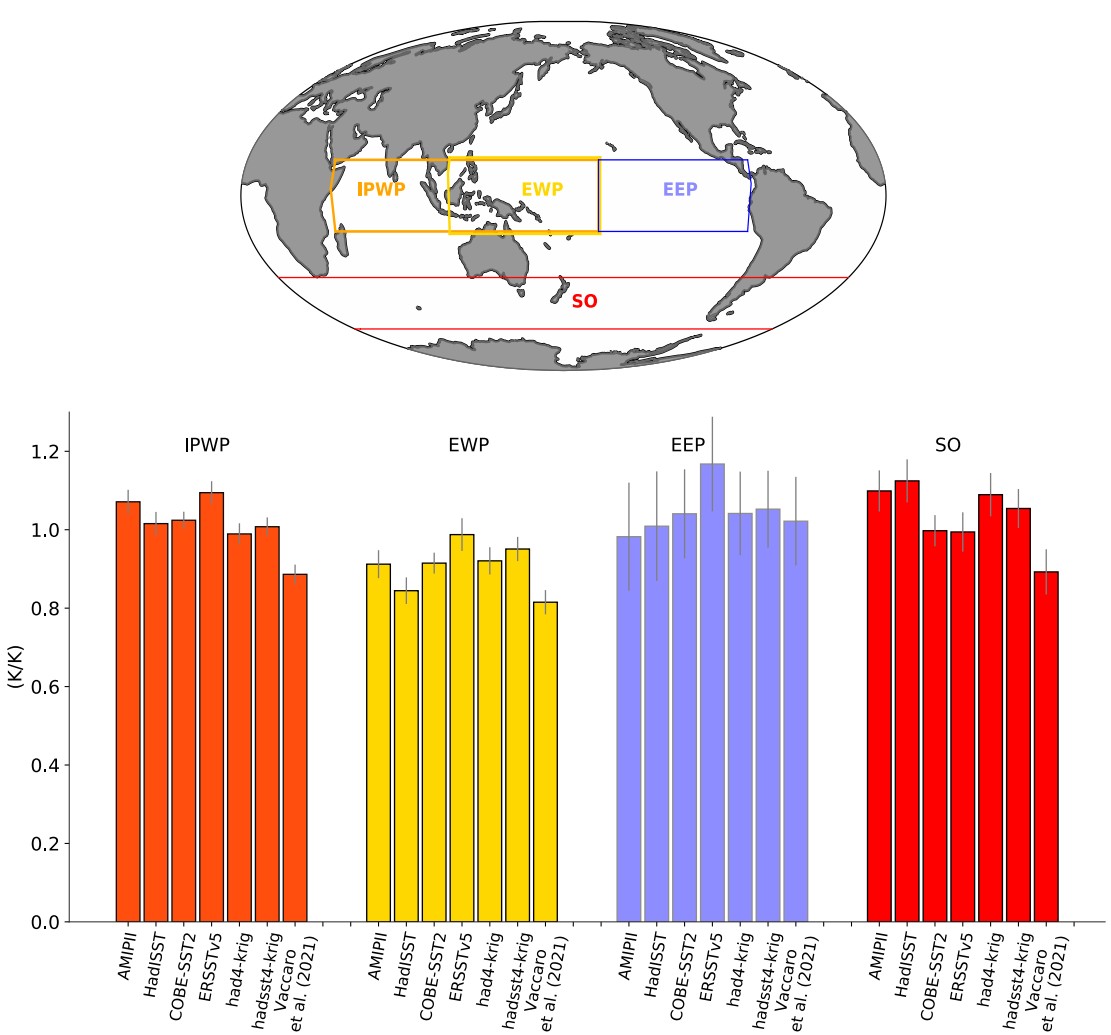

**Figure 5.** Top panel shows the selected regions. Bottom panel shows the local warming to the warming over 50°S-50°N (K/K) from 1871-2017 for the *observedSST-piForcing* simulations. The selected regions are Indo-Pacific Warm Pool (IPWP):15°S-15°N, 45°E-195°E; Equatorial West pacific (EWP):15°S-15°N, 110°E-195°E; Equatorial East pacific (EEP):15°S-15°N, 195°E-80°W and Southern ocean (SO):35°S-60°S. The error bars show ±1 standard error.

the pattern effect variations across datasets only to the IPWP warming, rather we find all regions show a positive correlation, with IPWP and SO showing a relatively stronger correlation than EWP and EEP (Figure 6).

To further investigate if any of the datasets bias the correlation, we calculate a range of correlation between the pattern effect and the relative warming by removing each dataset at a time. We find that over IPWP the Pearson coefficient correlation (r)

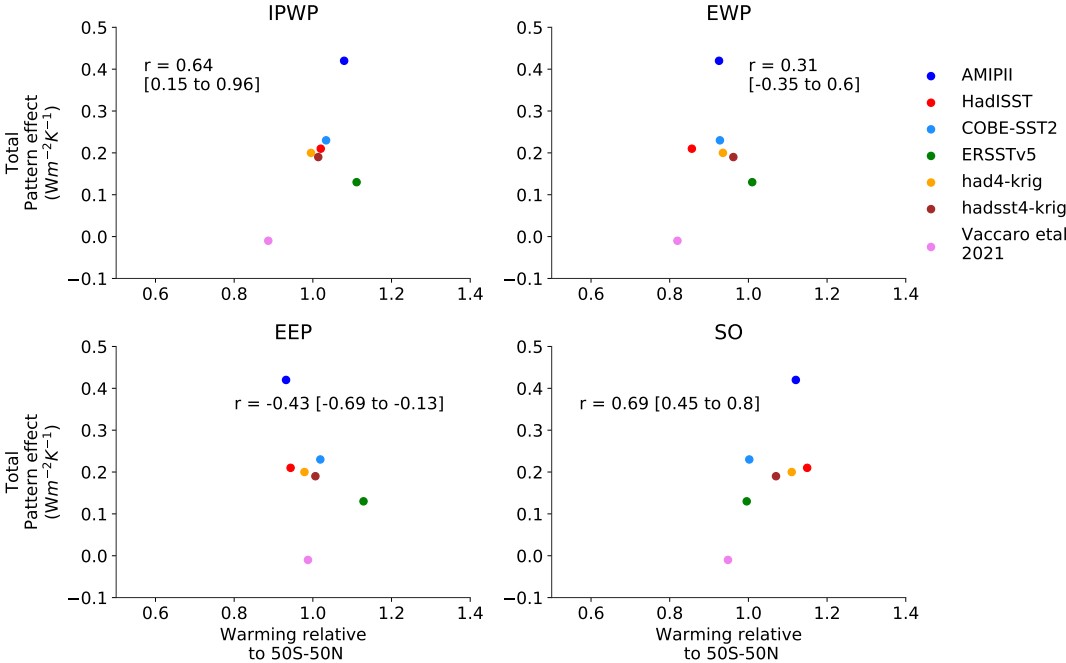

**Figure 6.** Relationship between the local warming in the selected regions to that over 50°S-50°N and the total pattern effect for the *observedSST-piForcing* simulations from 1871-2017. The selected regions are defined in Figure 5. The Pearson correlation coefficient (r) is shown in the respective panels. The ranges show the correlation coefficient by removing each dataset at a time.

ranges from 0.15 to 0.96: in absence of ERSSTv5 the correlation betters to 0.96 while without Vaccaro2021 it weakens to 0.15. Over EWP, EEP and SO the correlation ranges from -0.35 to 0.60, -0.69 to -0.13 and 0.45 to 0.80 respectively. Over EWP without ERSSTv5 the correlation coefficient improves to 0.60, however over EEP without AMIPII the correlation changes sign. Over SO the correlation coefficient is not sensitive to the datasets.

## 4.4 Temporal variation in pattern effect

Since the concept of pattern effect comes from the time evolving spatial structure of the surface temperature response, we regress $N$ against $T$ from the *observedSST-piForcing* simulations from 1871 to 1900 and then consecutively increase the regression length by a year to find the time varying nature of the feedback (Figure 7). Note that the final value of this feedback evolution is the 1871-2017 regression shown in Figure 4a, 4c and 4e. We find that the feedbacks from the *observedSST-piForcing* simulations have a large spread in the early period until 1940s, but during recent decades they agree more with each other. Unlike other datasets, feedback based on AMIPII starts to become more negative from the 1970s (Lewis and Mauritsen, 2021; Fueglistaler and Silvers, 2021). We find that feedback based on Vaccaro2021 also starts to drift relative to the rest from the 1970s (Figure 7a), but in the opposite direction. It is apparent that the SW-cloud feedback governs the evolution of the net

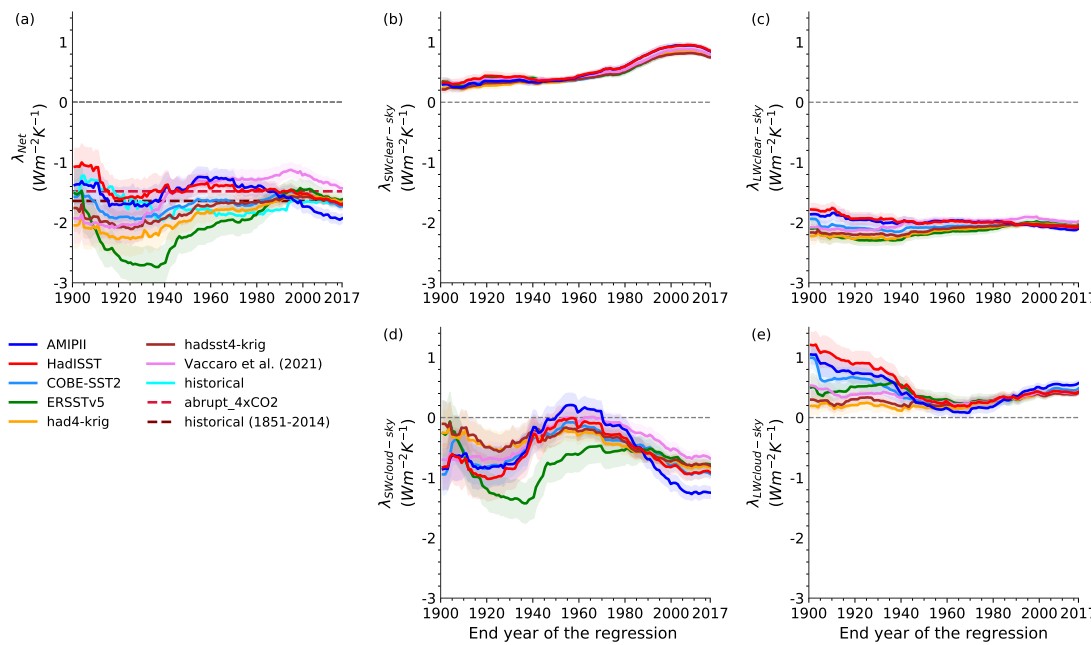

**Figure 7.** Net feedback (a), shortwave (SW)(b,d) and longwave (LW)(c,e) components of feedbacks obtained by regressing change in TOA radiative fluxes against change in surface temperature from 1871 to 1900 and then consecutively incrementing the regression length by a year from the *observedSST-piForcing* simulations. The shades represent the $\pm 1$ standard error while the dashed lines in (a) shows $\lambda_{4xCO2}$ and $\lambda_{hist}$. The cyan in (a) shows the time varying $\lambda_{hist}$ from the *historical* simulation which denotes the forced pattern effect.

feedback as shown in previous studies (Andrews et al., 2018; Fueglistaler and Silvers, 2021) and primarily responsible for the behaviour of feedbacks. Additionally, the spread in the early period is partly associated with the LW-cloud feedback (Figure 7e).

Another way to visualize the evolution of net feedback and its components is based on the regression of $N$ against $T$ in the
215 sliding 30-year windows, shown in Figure A3. We find that the feedback from all the *observedSST-piForcing* simulations is becoming less negative and consequently a smaller pattern effect in the most recent decade (Figure A3a).

### 4.5 An updated estimate of ECS

We now illustrate how ECS constrained based on the instrumental record reported in AR6 is updated when we account for the dataset dependency (Figure 8). We can integrate the pattern effect ($\Delta\lambda$) in the linear energy budget as, ECS $= -F_{2x}/(\lambda + \Delta\lambda)$.
220 In this section, note that we consider the best estimate while the uncertainty (standard deviation) and ranges denotes the 5-95% confidence intervals. We apply the energy imbalance anomaly between 1850-1900 and 2006-2019 from AR6 as $\Delta N = 0.59 \pm 0.35$ Wm$^{-2}$, the change in surface temperature as $\Delta T = 1.03 \pm 0.20$ K, the historical radiative forcing as $\Delta F = 2.20$ [1.53 to 2.91] Wm$^{-2}$ and the ERF for doubling of $CO_2$ as $F_{2x} = 3.93 \pm 0.47$ Wm$^{-2}$. We account for the correlated uncertainties

between the $F_{2x}$ and the ERF for the well-mixed greenhouse gases. We infer the estimate of ERF of well-mixed greenhouse
gases from AR6 as $1.73 \pm 0.29$ Wm$^{-2}$. The best estimate is obtained from the forcing data available at IPCC AR6 GitHub
repository. We deduced the uncertainty by taking the 5-95% ranges of ERF of the well-mixed greenhouse gases from *Figure
7.6* of AR6 report and adding them in quadrature.

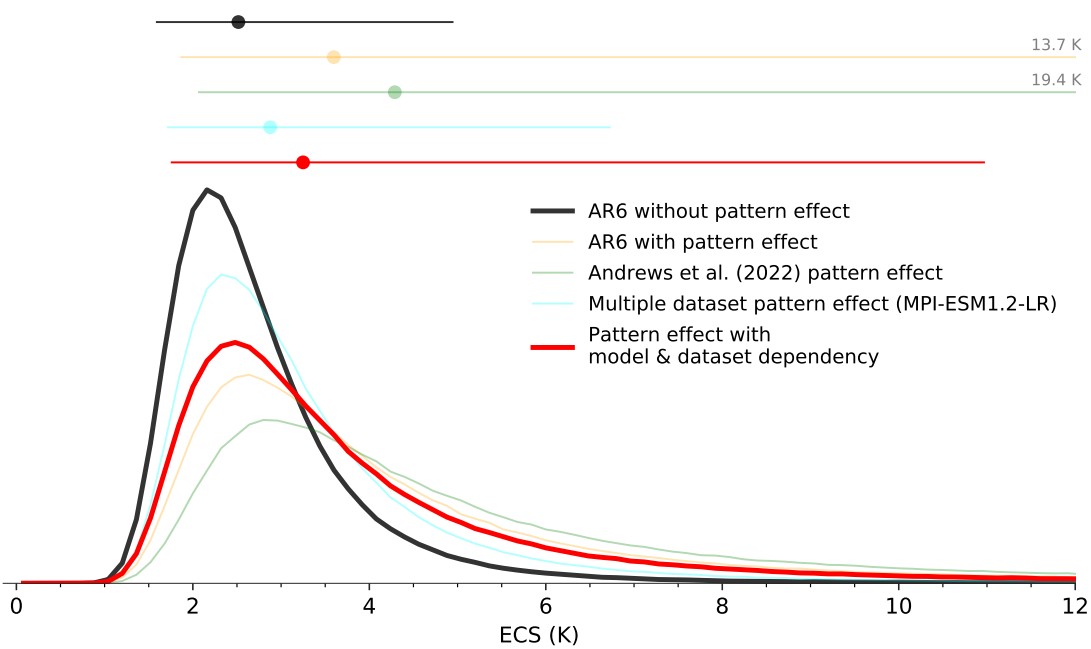

**Figure 8.** Probability distributions for ECS constrained based on the instrumental record of historical warming as per IPCC AR6 and updated
with pattern effect estimates as indicated by the legend. AR6 without pattern effect estimate is the baseline for all the other estimates.

Employing these values within the linear energy budget framework we reproduce the constrained ECS estimate to 2.5 K [1.6
to 4.9 K] (Figure 8). To compare the adjusted ECS derived based on our dataset dependent pattern effect, we first apply the
pattern effect used in the AR6, 0.50 Wm$^{-2}$K$^{-1}$ [0.0 to 1.0 Wm$^{-2}$K$^{-1}$]. This updates the ECS to 3.6 K [1.9 to 13.7 K]. When
we apply our dataset dependent pattern effect estimate - mean of $\Delta\lambda$ deduced from each dataset, 0.20 Wm$^{-2}$K$^{-1}$ [0.01 to 0.39
Wm$^{-2}$K$^{-1}$]), the ECS is adjusted to 2.9 K [1.7 to 6.7 K] (Figure 8). As expected this drops the best estimate as well as restricts
the ECS range owing to the smaller pattern effect strength and uncertainty. However, the adjusted ECS with this pattern effect
includes only the uncertainty due to different boundary conditions which are based on the observed-reconstructed datasets and
may be biased by the model. Andrews et al. (2022) estimated pattern effect considering AMIPII and HadISST datasets. Here,
we apply their pattern effect based on AMIPII only as they had *amip-piForcing* simulations from more models. They estimate
the pattern effect to be (0.70 Wm$^{-2}$K$^{-1}$ [0.23 to 1.17 Wm$^{-2}$K$^{-1}$]) which when applied lifts the AR6 ECS to 4.3 K [2.1 to
19.3 K]. However, this includes only the model uncertainty.

We therefore update the instrumental record constrained ECS from AR6 report with the combined pattern effect which includes the pattern effect based on the dataset dependency as well as the intermodel spread from Andrews et al. (2022). We assume that the uncertainties from model- and dataset- dependencies are independent. We find the combined pattern effect estimate to be 0.37 Wm$^{-2}$K$^{-1}$ [-0.14 to 0.88 Wm$^{-2}$K$^{-1}$]. Our calculations for the combined estimate considers only the AMIPII based pattern effect estimates from Andrews et al. (2022). The combined estimate is obtained by subtracting the mean of ECHAM6.3 and MPI-ESM1.2-LR estimate in Andrews et al. (2022) from the sum of our estimate and their multi-model mean estimate. The uncertainty range in the combined estimate is deduced by adding in quadrature the standard deviations from our dataset dependent estimate and their multi-model estimate. Thus, our calculation also accounts for the weaker pattern effect in ECHAM6.3 and MPI-ESM1.2-LR than in the multi-model mean. While accounting for the weaker pattern effect the assumption is that ECHAM6.3/MPI-ESM1.2LR is different from all the other models in all datasets as in AMIPII. However, one can argue this assumption. We infer from Andrews et al. (2022) and check this. We show in Figure A4 that not only ECHAM6.3/MPI-ESM1.2LR but also other models though produce stronger pattern effect has a similar difference in pattern effect estimates based on AMIPII and HadISST datasets as in ECHAM6.3/MPI-ESM1.2LR. Nevertheless, when we apply the combined pattern effect, we find that the constrained ECS based on historical warming is adjusted to 3.2 K [1.8 to 11.0 K] (Figure 8) which is slightly smaller and better constrained than in the instrumental record estimate of the IPCC AR6.

## 5 Conclusions

The observed historical warming provides an opportunity to estimate Earth's equilibrium climate sensitivity. However, our ability to constrain ECS based on it is limited both by the uncertainty in the aerosol cooling and the strength of pattern effects. Pattern effects temporarily dampen transient global warming and so factoring this leads to larger and less constrained ECS estimates. Recent studies have estimated the model-dependence of the pattern effect strength using a range of models with prescribed SSTs primarily from the AMIPII observed-reconstructed dataset. Here we instead investigated the dataset-dependence using a single model with seven different datasets. The resulting spread is substantial, although smaller than that among different models, and it turns out that the pattern effect estimated from the AMIPII dataset is by far the largest suggesting that earlier studies may have overestimated the pattern effect. However, we reiterate that any of the SST dataset could be a possible path Earth could have taken.

The datasets differ due to the measurement sources, methods applied to merge datasets and the infilling techniques. Depending on the applied datasets, we find that the total pattern effect varies from -0.01 ± 0.09 Wm$^{-2}$K$^{-1}$ to 0.42 ± 0.10 Wm$^{-2}$K$^{-1}$ over 1871-2017, while the mean across the datasets is 0.20 Wm$^{-2}$K$^{-1}$ [0.01 to 0.39 Wm$^{-2}$K$^{-1}$]. The mean unforced pattern effect across the dataset is generally small although it ranges from $-0.17 \pm 0.11$ Wm$^{-2}$K$^{-1}$ to $0.26 \pm 0.11$ Wm$^{-2}$K$^{-1}$. As expected, differences in the pattern-effect is primarily attributed to differences in the cloud radiative effects. While the estimates from the 1970s until present are less dataset dependent, the major disparities originates in the early period and is driven by cloud feedback. By assuming the variations in model- and dataset-dependencies are independent we then estimate a combined pattern effect of 0.37 Wm$^{-2}$K$^{-1}$ [-0.14 to 0.88 Wm$^{-2}$K$^{-1}$]. Taking global warming, radiative forcing and imbalance from

AR6, this results in a historical warming constrained ECS estimate of 3.2 $K$ [1.8 to 11.0 $K$], which is better constrained than that in the AR6 report as a result of the slightly weaker mean pattern effect.

In the community currently the prevailing understanding is that the strength of the pattern effect is related to the temperature response over the Indo-Pacific Warm Pool. However, here we find comparative correlation between the pattern effect strength and the relative warming trends over all the Indo-Pacific Warm Pool, Equatorial West Pacific, Equatorial East Pacific and Southern Ocean to that over 50°S-50°N across the datasets. Thus, we are unable to identify specific regions that could govern the strength of the pattern effects and it is difficult to co-locate patterns of surface temperature change and the strength of pattern effect between the datasets.

In the current study we had to assume that variations in the pattern effect as estimated among the models and across the datasets are independent of each other in order to provide a combined estimate of the pattern effect. However, one could raise a concern that a single model will not completely display the dataset dependency. In particular, the model we used here is correlated less well with the observed shortwave anomalies in Loeb et al. (2020). Therefore, we have decided in extension to conduct a multi-model and multi-dataset intercomparison project to address this outstanding concern.

## Appendix A

Table A1: List of observed SST datasets applied to MPI-ESM1.2-LR in this study.

| Data set | Description |
|---|---|
| HadISST1 (Rayner et al., 2003) | Resolution: 1 × 1 degree From 1871-1995, the SST data is taken from the Comprehensive Ocean Atmosphere Dataset (COADS). From 1981 onwards – AVHRR satellite surface skin temperature are also used in conjunction. It applies a two-stage reduced-space optimal interpolation procedure along with superposition of quality-improved gridded observations onto the reconstructions to restore local details. |

| Data set | Description |
|---|---|
| AMIPII (Hurrell et al., 2008) | Resolution: 1 × 1 degree<br>Standard SST data used for amip-piForcing (CMIP6/CFMIP) simulations. It is a merged product based on HadISST1 and OI.v2 SST fields. Before Nov 1981, it uses HadISST1 while afterwards uses OI.v2 SST fields. The merging is done by first adding the HadISST1 anomalies relative to its own based period to OI.v2 climatology for the same base period and then adjusting that eliminates outliers and produce physically realistic SST values. The re-basing and adjustments changes relative temperature values prior to 1981 as well. OI.v2 also employs same AVHRR source as HadISST1 but applies different techniques of assimilation and bias correction. Here, further adjustments are made to monthly SST fields to preserve seasonal cycle amplitude when interpolated to daily timescale. |
| COBE-SST2 (Hirahara et al., 2014) | Resolution: 1 × 1 degree<br>SST field constructed as a sum of a trend, interannual variations and daily changes, using in situ SST and sea ice concentration observations (ship+buoy). Satellite observations are adopted for the purpose of reconstruction of SST variability over data-sparse regions. Employed infilling method is claimed to be superior to direct use of optimal interpolation when data is sparse. |
| ERSSTv5 (Huang et al., 2017) | Resolution: 2 × 2 degree<br>This dataset is based on in-situ measurements from ships and buoys but upto 2010 the ship SST values are based on HadNMAT2 night time marine air temperature data. It applies OIv2 dataset same as AMIPII but the infilling method is different. The colder temperature during 1900-1980 relative to other datasets is associated with higher ship SST bias correction during 1880-1940s and 1950-1960s. |
| had4-krig-v2-0-0 (Cowtan and Way, 2014) | Resolution: 5 × 5 degree<br>It is based on HadCRUT4.6 temperature data (Morice et al., 2012) where the kriging method is applied for producing spatially complete data. The SST data is a krigged version of HadSST3 which is based on in-situ measurements from ships and buoys. |
| had4sst4-krig-v2-0-0 (Cowtan and Way, 2014) | Resolution: 5 × 5 degree<br>Same as had4-krig-v2-0-0, but the SST data is a krigged version of HadSST4. |

| Data set | Description |
|---|---|
| Vaccaro2021 (Vaccaro et al., 2021) | Resolution: 5 × 5 degree It is based on HadCRUT4.6 temperature data (Morice et al., 2012). Here, a Gaussian graphical model called as Graphical Expectation-Maximization algorithm is applied to produce spatially complete estimate of HadCRUT4.6 data. This infilling method gives improved estimates of missing values compared to other methods such as kriging (Vaccaro et al., 2021). This data produces realistic reconstruction of past climates like 1877-1878 El-Nino and stronger historical warming trends than the ones which are not interpolated. This dataset comes with 100 ensemble members of temperature reconstructions and we have applied the median temperature reconstruction. |

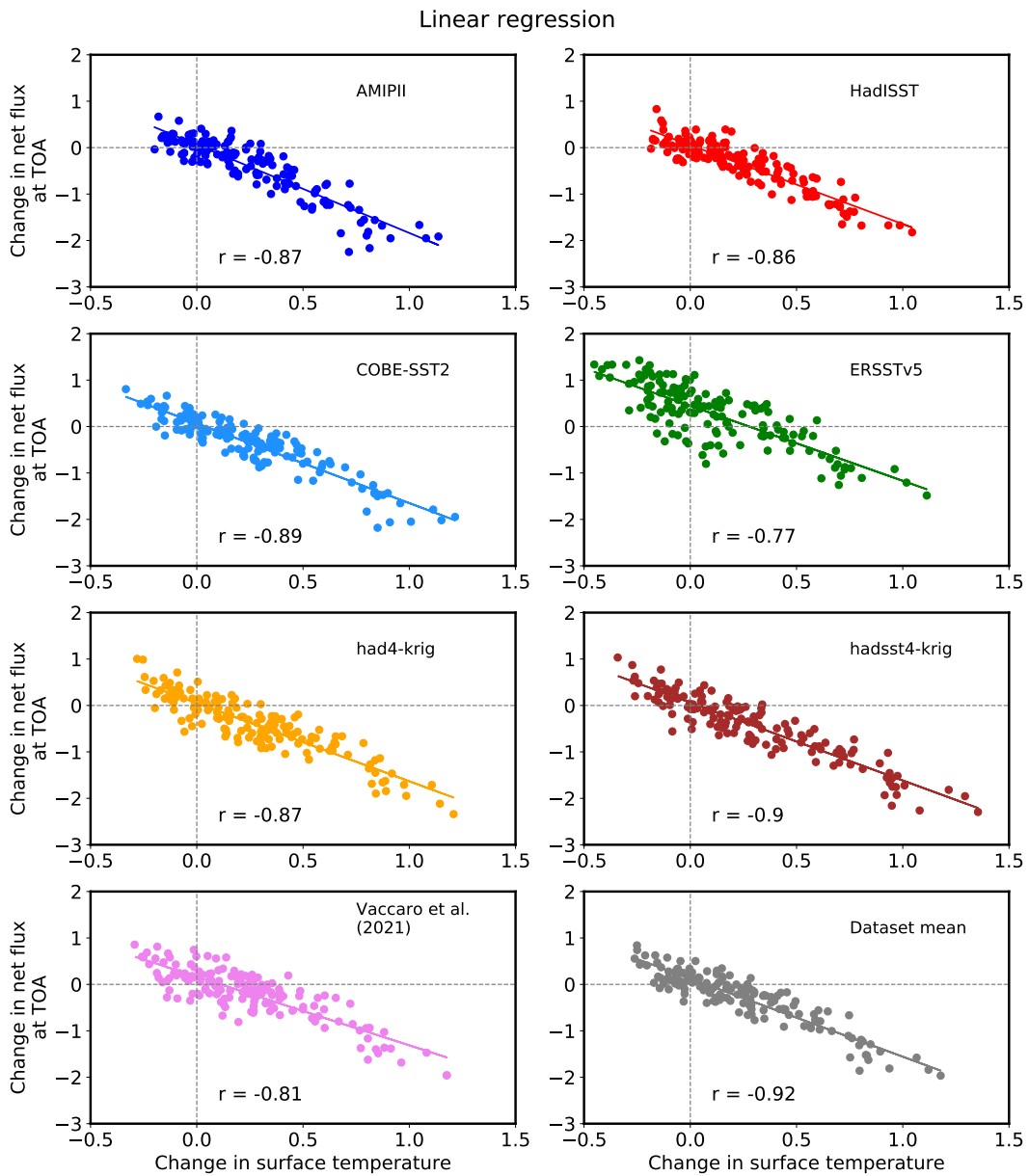

**Figure A1.** Linear regression between the change in global-annual mean net downward radiation flux at TOA against the surface temperature change (Gregory et al., 2004) for the *observedSST-piForcing* simulations. The changes are relative to the 1871-1900 mean. The Pearson correlation coefficient (r) is shown for each panel.

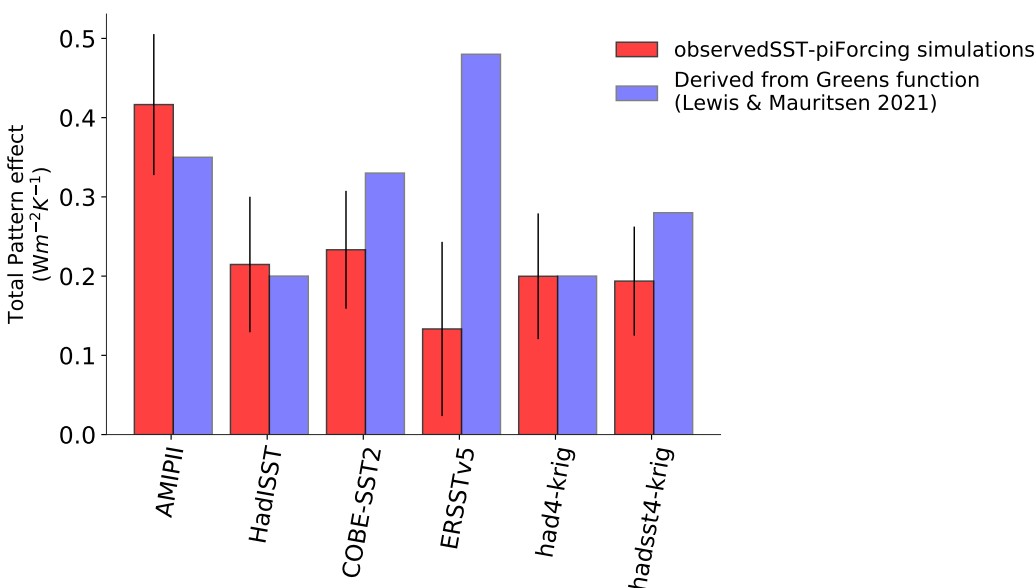

**Figure A2.** Comparison of total pattern effect estimated from the *observedSST-piForcing* simulations (as in Figure 4a) and inferred from Lewis and Mauritsen (2021) for the in-common SST datasets.

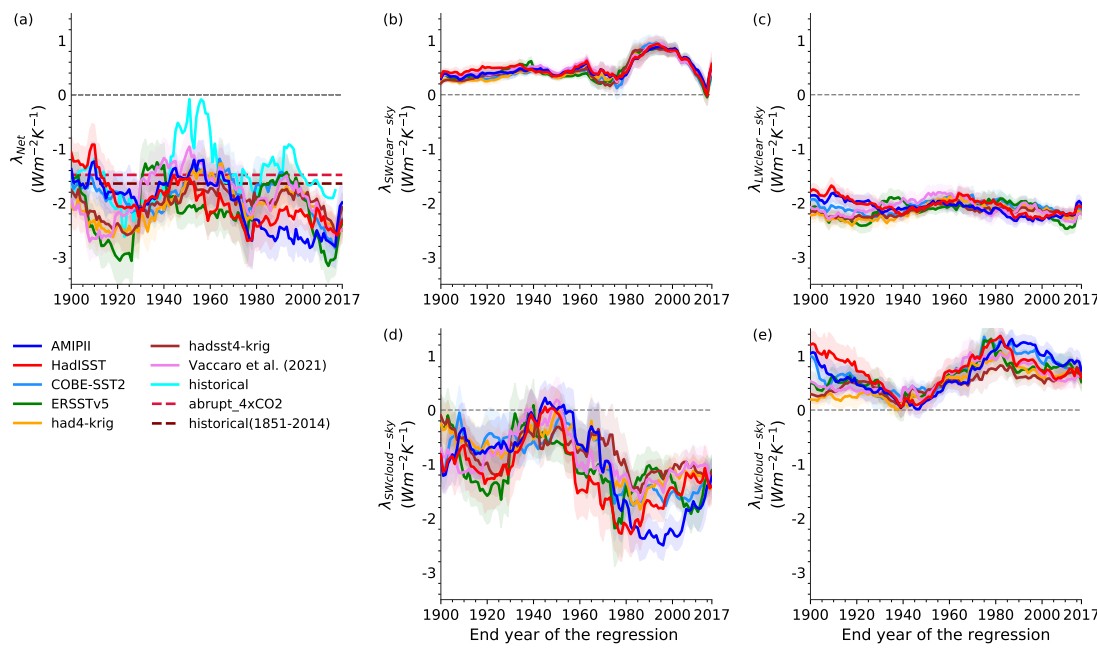

**Figure A3.** Net feedback obtained by regressing change in TOA radiative fluxes against the change in surface temperature with a sliding 30-year period from 1871 to 2017 from the *observedSST-piForcing* simulations. The shades represent the ± 1 standard error while the dashed lines in (a) shows $\lambda_{4xCO2}$ and $\lambda_{hist}$. The cyan in panel (a) shows net feedback from *historical* simulation for MPI-ESM1.2-LR.

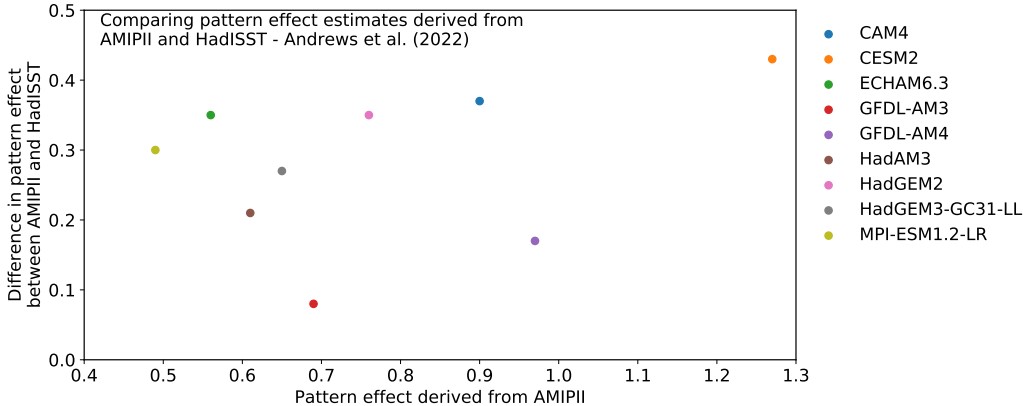

**Figure A4.** Difference in the total pattern effect estimates derived from AMIPII and HadISST datasets for a given model against its total pattern effect estimate based on AMIPII. Shown for all models (in legend) from Andrews et al. (2022).

*Code and data availability.* The source code for MPI-ESM1.2-LR is available through https://mpimet.mpg.de/en/science/models/mpi-esm. MPI-ESM1.2-LR model output for the *piControl, abrupt4xCO2 and historical* simulations are freely available from the Lawrence Livermore National Laboratory, World Climate Research Programme (WCRP), 2019 (https://esgf-node.llnl.gov/search/cmip6/). The observed-reconstructed SST datasets HadISST1, AMIPII, COBE-SST2, ERSSTv5, had4-krig-v2-0-0, had4sst4-krig-v2-0-0 and Vaccaro2021 are available from https://www.metoffice.gov.uk/hadobs/hadisst/, https://pcmdi.llnl.gov/mips/amip/amip2/#data, https://psl.noaa.gov/data/gridded/data.cobe2.html, https://www.ncei.noaa.gov/products/extended-reconstructed-sst, https://www-users.york.ac.uk/ kdc3/papers/coverage2013/series.html and https://doi.org/10.5281/zenodo.4601616, respectively. Data associated with the figures is publicly available at https://doi.org/10.5281/zenodo.7106446

*Author contributions.* The initial project idea was conceived by TM and is developed during AM and TM discussion. AM obtained the SST datasets and worked on them to perform the simulations. AM did the analysis. AM and TM discussed the results and wrote the manuscript.

*Competing interests.* The authors declare no conflict of interest.

*Acknowledgements.* The authors thank Tim Andrews and the anonymous reviewer for the comments and suggestions that helped advance this study. AM thanks Martin Renoult and Kyle Armour for the discussions. This work was supported through the funding from the European Research Council (ERC) Grant agreement 770765 and the European Union's Horizon 2020 research and innovation program projects CONSTRAIN and NextGEMS (Grant agreements No.820829 and No. 101003470). The simulations and the analysis was enabled by the resources provided by the Swedish National Infrastructure for Computing (SNIC) at Stockholm University partially funded by the Swedish Research Council through grant agreement 2018-05973.

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
