# Peer review of "Better constrained climate sensitivity when accounting for dataset dependency on pattern effect estimates"

_EGUsphere, 2022_

## Referee Comment (RC1)

The authors present a new estimate of how much the "pattern effect" affects equilibrium climate sensitivity (ECS) inferred from 1871—2017 climate change. Many pattern-effect studies use the AMIPII sea-surface temperature (SST) dataset but there are unresolved inter-dataset differences in SST evolution. The submission shows how, in a single climate model, different SST datasets affect estimated ECS_hist. The paper is in scope and takes a sensible approach. Despite my review's length, I picked minor revisions because I expect the main conclusions to be robust – with the caveat that all estimates of ECS_hist are expected to have issues.

Before supporting publication, I ask that the authors address a series of comments on several themes below, then end with minor comments.

I strongly suggest that panel Fig. A1(a) goes into the main manuscript. It allows identification of which periods matter for ECS. I was interested in WWII when ERSST and HadSST-based datasets have very different corrections, plus recent decades for which we have satellite-based constraints. Different communities will care about such results which are hard to interpret from Fig. 7(a).

**1. Discussion and results to contextualise this paper**

I think your discussion of the causes of historical pattern effects is insufficient (e.g. "could be either externally forced or could be internal variability driven"). As I understand it, your ECS_hist correction is only valid if the pattern effect is internal variability and/or non-CO2 forced. If it's CO2 forced, as proposed in Seager et al. (2019, doi: 10.1038/s41558-019-0505-x), then your correction would be invalid. Please explicitly discuss.

I believe you could also derive time-varying $\lambda_{hist}$ as in Figs. 7(a) and A1(a) from the historical runs. Does that suggest a role for a time-varying forced pattern effect in this model?

It also seems obvious to compare to prior work. Can you reproduce Fig. 7(a) and Fig. A1(a) with the Lewis & Mauritsen (2021) CAM5.3 Green's function results for the in-common SST datasets?

Both tests could go in the appendix with brief main-text commentary, unless results are particularly interesting.

**2. Method clarifications**

Some methodology details confused me.

P5L108 paragraph. Did you remove piControl drift? Was preindustrial volcanism included in the piControl runs? Do these factors change $N$ and therefore ECS_hist?

For the standard error from SST datasets, is that just the Gaussian standard error on the mean, treating each SST result as independent?

How do you combine the errors in Sec. 4.5? I followed AR6 Fig. 7.6 to Table 7.SM.14 and the cited notebook (https://github.com/chrisroadmap/ar6/blob/main/notebooks/100_chapter7_fig7.6.ipynb). I don't see where error combination method is stated. Please clarify.

**3. Robustness tests and some extra detail**

How linear are things?

Can you show (perhaps appendix?) annual $N$ versus $T$ by SST dataset? Can you check your lambda regression calculations with something more robust than OLS to outliers (e.g. Theil-Sen)? The results would help say something about robustness.

Long-term linear fits to SST(t) could be misleading. What if Fig. 3, 5 & 6 analysis were repeated with e.g. LOWESS or differences between start and end periods?

Are there periods (e.g. from the 30-year regressions) where the Fig. 5/6 regions seem more predictive of $\lambda$? Only interesting results need to go into the paper or appendix. You might find times or locations that are particularly impactful.

For the Figs. 5/6 regional analysis – did you try any combinations e.g. a Pacific west-east difference? Is there no additional info from a larger Sc/trade Cu region? I understand you should limit paper content, but it would be nice to know you checked at least Pacific ascent minus descent region differences given the number of papers that have highlighted the east-west gradient.

You report narrowed ECS uncertainty but to some it may be counterintuitive that adding SST source uncertainty shrinks the final errors. Your title emphasises the reduced uncertainty – but somewhere it should be stated that this just comes from the smaller ECS thanks to how the division-by-lambda works. Spelling this out would be helpful, as would mentioning some extra caveats that I did not see:
- Are SST dataset lambdas treated as independent in the standard error calculation? If yes, are your standard errors too small since they're not actually independent?
- Do you use model spread as an error? If yes, please add the usual comments about ensemble of opportunity etc and how that affects interpretation.
- You said that you "*had to assume that variations in the pattern effect as estimated among the models and across the datasets are independent of each other*". It seems likely there is a correlation though, justifying your suggested next steps in the final sentence. You could do error propagation with assumed correlations between model and SST errors and a sentence or two could then say "strong correlation (r=whatever) does/doesn't greatly affect these conclusions…".

Basically your headline is "reduced uncertainty" and I'd like extra clarity on exactly what you're claiming.

**Minor/typos**
Suggest extending subscripts to be absolutely clear, specifically "ECS_hist" (or "ECS_amip") to discriminate as you do with your $\lambda$ subscripts.

P1L11 – 95 percentile -> 95[th] percentile

P1L19—20: "*To know what is required to meet the Paris Agreement goal it is imperative to better quantify and understand the rate of global warming*"
You aren't really studying warming rates. Maybe something like "…it is imperative to better quantify and understand the ultimate amount of warming in response to a given forcing."

P2 lines messed up but between 25 and 30:
"*It can be framed as N = F +$\lambda$T, where N is the planetary energy imbalance which is generally measured as the net downward radiative flux at the top-of-the-atmosphere (TOA), F is the external radiative forcing, measured as the effective radiative forcing (ERF)*"
I think you mean "defined" rather than "measured", both for TOA flux (even CERES is pinned to Argo etc for the mean) and ERF is clearly not measured either.

P2 lines and equation:
The equation uses $\Delta T$ and the text uses $T$ both to denote changes in $T$. The same for $N$ and $F$. I suggest using $\Delta T$, $\Delta F$ and $\Delta N$ consistently to denote changes.

P2L32—33:
"*To reconcile this discrepancy the community looked into the concept of pattern effect which is not taken into account in the traditional energy balance framework.*"
This misses how changes in $T$ data have reduced the apparent differences. Both Lewis & Curry and Otto et al. used HadCRUT4. Pick some relevant citation – a recent e.g. is Clarke & Richardson (2021, doi: 10.1029/2020EA001082) whose Table 2 shows how 1850—2019 ECS_hist estimates would increase relative to HadCRUT4. Cowtan & Way infilling is worth about +10 %, while Berkeley

Earth's more substantial improvements (more stations, infilling, improved sea ice treatment) give +17 % to ECS_hist. The authors could persuade me otherwise, but updated global temperature datasets seem both important context for a study based on inter-dataset issues.

P2L42: "*Atmosphere-Ocean General Circulation Model (AOGCM) simulations show that the long term climate response (response to abrupt4xCO2) resembles a temperature pattern similar to 'El-Nino Southern Oscillations' (ENSO)*"
This could be interpreted to mean that models have shown/proven the "real" long-term pattern and should be rephrased to something like: AOGCMs' *simulated* pattern resembles ENSO. I think you wanted to make this point in P3L47 when you said "assumed correct", but I wasn't sure. Change from:
"*…leads to more stabilizing feedback while an assumed correct opposite future temperature distribution would lead to a less stabilizing feedback*"
To something like:
"*…leads to more stabilizing feedback while the simulated future temperature distribution would lead to a less stabilizing feedback*"

Figure 1 caption: are your fixed-SST simulations really "AOGCM" which you've defined as "atmosphere ocean GCMs"? I interpreted AOGCM as including coupling to an ocean model.

P3L57—58: "*Such estimates are model dependent, nevertheless most models yield a dampening pattern effect based on the AMIPII dataset.*"
Alone, "dampening effect" could simply mean negative feedback in general. I assume you mean dampening relative to the long-term abrupt4xCO2 pattern? If yes, please correct.

Fig. 3 – second row first column figure is labelled "HadSST" rather than "HadISST"

P5L113—115: "*For λhist, we take the changes in the global annual mean N and T from 1851-2014 of historical simulation relative to the mean of 1851-1900 of the same. Since, there are 10 ensemble members present in the historical simulation, we use the mean of the ensembles in our calculation*"
Can you rephrase? I think this is saying you took the mean of 1851-2014 minus the mean of 1841-1900 but am not certain. Also, did you mean "ensemble mean" instead of "mean of ensembles"?

P8L142: is this the mean and standard error of the fits to each dataset, calculated using the standard Gaussian assumption, or something else?

P8L145: rogue decimal point in error: 0.0.25

P8L151: typo "expectced" -> "expected"

P8L161: typo gap in 0 .59

P10Fig6: Are $y$ axis units W m$^{-2}$ K$^{-1}$? Please add axis labels or explain why not.

P11L195—196: "*while the uncertainty is deduced from Figure 7.6 of this report.*" - "this report" could mean your paper, whereas I infer you mean "AR6 Figure 7.6"

P12L204: typo "bt" -> "by"

P14L245: Table A1 says that AMIPII = HadISST until Nov 1981, is that correct? If yes, then what are the slight differences in Fig. 1? If so, then is it surprising that there are such big, consistent differences in feedbacks in Fig. 7(a) when you've got multiple runs with only land/atmosphere variability?

---

## Author Comment (AC1)

**Review of "Better constrained climate sensitivity when accounting for dataset dependency on pattern effect estimates" by Modak and Mauritsen.**

**Note- Our responses to the reviewer's comments are in red color text.**

The authors present a new estimate of how much the "pattern effect" affects equilibrium climate sensitivity (ECS) inferred from 1871—2017 climate change. Many pattern-effect studies use the AMIPII sea-surface temperature (SST) dataset but there are unresolved inter-dataset differences in SST evolution. The submission shows how, in a single climate model, different SST datasets affect estimated ECS_hist. The paper is in scope and takes a sensible approach. Despite my review's length, I picked minor revisions because I expect the main conclusions to be robust – with the caveat that all estimates of ECS_hist are expected to have issues.

Before supporting publication, I ask that the authors address a series of comments on several themes below, then end with minor comments.

I strongly suggest that panel Fig. A1(a) goes into the main manuscript. It allows identification of which periods matter for ECS. I was interested in WWII when ERSST and HadSST-based datasets have very different corrections, plus recent decades for which we have satellite-based constraints. Different communities will care about such results which are hard to interpret from Fig. 7(a).

We thank the reviewer for appreciating the approach of this study and for the constructive comments that helped improve the manuscript. The point by point responses are provided below.

Thanks for the suggestion on Figure A1(a). To update ECS estimates based on the historical warming, the entire long-term period is generally applied, in this case 1871-2017. In Figure 7 we show how the long-term slope evolves when we increase the regression length by a year derived from different SST datasets. Hence, we find it relevant for this study and decide to retain Figure 7 in the main text. We agree that the sliding 30-year evolution of climate feedback is useful as it helps to identify how the 30-year feedback varies and can be used to compare the datasets. However, we do not use the 30-year periods to update ECS where the start and end year of the regression changes, and so is not central to our work. Hence, we decide to keep it in the supplementary.

**1. Discussion and results to contextualise this paper**

I think your discussion of the causes of historical pattern effects is insufficient (e.g. "could be either externally forced or could be internal variability driven"). As I understand it, your ECS_hist correction is only valid if the pattern effect is internal variability and/or non-CO2 forced. If it's CO2 forced, as proposed in Seager et al. (2019, doi: 10.1038/s41558-019-0505-x),

then your correction would be invalid. Please explicitly discuss.

This is an interesting point, although we are not entirely sure what the reviewer thinks is invalid. For a given model the forced pattern effect can be derived from an ensemble of historical simulations. This pattern effect is the result of all forcing agents applied in the respective historical simulation. Taking the same model and running it with prescribed SST patterns from reconstructions then gives us a model estimate of the total pattern effect. The difference may then be interpreted to be due to internal variability.

We now clearly state and discuss in the revised manuscript "We call it "total" to highlight the fact that the *observedSST-piForcing* AGCM simulations encapsulates the radiative effect of spatial distribution of temperature which depends on the all external forcings during the historical period as well as internal variability (Gregory et al., 2019; Seager et al., 2019; Watanabe et al., 2020; Lewis and Mauritsen, 2021). On the other hand, for a given model the forced pattern effect can be derived from an ensemble of historical simulations. This pattern effect is the result of all forcing agents applied in the respective historical simulation. The difference between the "total" and "forced" could then be interpreted to be due to internal variability (discussed in next section). It is worth mentioning that recent studies (e.g. Seager et al., 2019) while interpreting the temperature gradient in the equatorial Pacific proposed that rising greenhouse gases is the cause for the observed temeprature gradient. However, other studies supports natural forcing internal variability are equally possible causes (e.g. Gregory et al., 2019; Watanabe et al., 2020; Olonscheck et al., 2020).

I believe you could also derive time-varying $\lambda_{hist}$ as in Figs. 7(a) and A1(a) from the historical runs. Does that suggest a role for a time-varying forced pattern effect in this model?

Thanks for the suggestion. In the revised manuscript, we have now added the time varying climate feedback from the historical simulation Figure 7(a) and A3(a). We derive it from the mean of the 10 ensemble members of historical simulations. For a given model the forced pattern effect can be derived from an ensemble of historical simulations. This pattern effect is the result of all forcing agents applied in the respective historical simulation. So, yes, time-varying $\lambda_{hist}$ from the historical simulations is the forced pattern effect.

It also seems obvious to compare to prior work. Can you reproduce Fig. 7(a) and Fig. A1(a) with the Lewis & Mauritsen (2021) CAM5.3 Green's function results for the in-common SST datasets? Both tests could go in the appendix with brief main-text commentary, unless results are particularly interesting.

Thanks. Instead of reproducing Figure 7(a) and Figure A1(a), in the revised manuscript, we add a figure (also shown below) which compares the total pattern effect estimated based on the *observedSST-piForcing* simulations of this study and the total pattern effect inferred from Table-2 of (Lewis and Mauritsen, 2021) which are derived based on CAM5.3 Green's function. We find that the pattern effect is substantially different between the two in the case of ERSSTv5, COBE-SST2, hadSST4-krig and had4-krig.

In the revised manuscript, we add "We inferred the total pattern effect from (Lewis and Mauritsen, 2021) for the in-common SST datasets which they calculated based on CAM5.3 Green's function (Figure A1). We find that the estimates of total pattern effect are substantially different from our estimates for some of the SST datasets."

[Figure]

Figure 1

**2. Method clarifications**

Some methodology details confused me.

P5L108 paragraph. Did you remove piControl drift? Was preindustrial volcanism included in the piControl runs? Do these factors change N and therefore ECS_hist?

Thanks. We estimate the changes relative to the last 500 years of piControl simulation and do not remove the piControl drift. We have cheked and find that removing the piControl drift instead does not change the estimate of ECS. Further, MPI-ESM1.2-LR has more than 10000 years spin-up *piControl* simulation and does not have a drift (Mauritsen et al. 2019).

MPI-ESM1.2 model do not have pre-industrial volcanic stratospheric aerosols prescribed in the spin-up and piControl simulations (Mauritsen et al. 2019).

For the standard error from SST datasets, is that just the Gaussian standard error on the mean, treating each SST result as independent?

We write in section *3. Model, datasets and experiments* "Unless otherwise specified, throughout the text, the displayed results are based on the mean of the 5 ensemble simulations while the uncertainty denotes the standard error from the ordinary least square (OLS) regression and the ranges are the 5-95$^{th}$ percentiles." We now add in section *4.5 An updated estimate of ECS* "IN this section, note that we consider the best estimate while the uncertainty (standard deviation) and ranges denotes the 5-95% confidence intervals."

How do you combine the errors in Sec. 4.5? I followed AR6 Fig. 7.6 to Table 7.SM.14 and the cited notebook (`https://github.com/chrisroadmap/ar6/blob/main/notebooks/100 _chapter7_fig7.6.ipynb`). I don't see where error combination method is stated. Please clarify.

We took the 5-95% ranges of ERF for $CO_2$ and other well-mixed greenhouse gases from Table 7.6, estimated the standard deviations and have added them in quadrature to obtain the uncertainty. In the revised manuscript, we add "We deduced the uncertainty by taking the 5-95% ranges of ERF of the well-mixed greenhouse gases from *Figure 7.6* of AR6 report and adding them in quadrature."

**3. Robustness tests and some extra detail**

How linear are things? Can you show (perhaps appendix?) annual N versus T by SST dataset? Can you check your lambda regression calculations with something more robust than OLS to outliers (e.g. Theil-Sen)? The results would help say something about robustness.

We include a figure in supplementary *Figure A1* that shows the OLS regression between annual N versus T by SST dataset. We redid the calculations with Theil-Sen regression and find that it considers the recent warming period as outliers, which is not particularly helpful. We do not include it in the manuscript. Both figures are shown below.

[Figure]

Figure 2

[Figure]

Figure 3

Long-term linear fits to SST(t) could be misleading. What if Fig. 3, 5 6 analysis were repeated with e.g. LOWESS or differences between start and end periods?

In agreement with Referee no. 2, we find a calculation of more relevance to feedback and patterns effects is the SST change per global-mean temperature change, rather than a trend in time. Hence, in the revised manuscript, we have updated the analysis of Figure 3, Figure 5 and Figure 6 by regressing dT(lat,lon,t) against global-mean dT(t) instead of temperature trends as a function of time. We find that the overall results do not change.

Are there periods (e.g. from the 30-year regressions) where the Fig. 5/6 regions seem more predictive of $\lambda$? Only interesting results need to go into the paper or appendix. You might find times or locations that are particularly impactful.

For the Figs. 5/6 regional analysis – did you try any combinations e.g. a Pacific west-east difference? Is there no additional info from a larger Sc/trade Cu region? I understand you should limit paper content, but it would be nice to know you checked at least Pacific ascent minus descent region differences given the number of papers that have highlighted the east-west gradient.

We find that this additional analysis is beyond the scope of the current study. This can be addressed in a different study.

You report narrowed ECS uncertainty but to some it may be counter intuitive that adding SST source uncertainty shrinks the final errors. Your title emphasises the reduced uncertainty – but somewhere it should be stated that this just comes from the smaller ECS thanks to how the division-by-lambda works. Spelling this out would be helpful, as would mentioning some extra caveats that I did not see:

In the revised manuscript in the abstract we add "... which as a result of the weaker pattern effect is slightly lower and better constrained than in previous studies" and in the conclusion section we add "which is better constrained than that in the AR6 report as a result of the slightly weaker mean pattern effect".

- Are SST dataset lambdas treated as independent in the standard error calculation? If yes, are your standard errors too small since they're not actually independent?

When we apply our dataset dependent total pattern effect estimate to update the instrumental record constrained ECS estimate from AR6, we consider $\lambda$ from the *observedSST-piForcing* simulations independent. We calculate the mean of $\Delta\lambda$ and the standard deviation is calculated as the deviation among the $\Delta\lambda$ from the *observedSST-piForcing* simulations.

In the revised we add more discussion.

We acknowledge in section *3. Model, datasets and experiments* that the measurement sources of some of the SST datasets from where they are constructed from is common.

- Do you use model spread as an error? If yes, please add the usual comments about ensemble of opportunity etc and how that affects interpretation.

Yes, we apply the model spread from (Andrews et al., 2022) when we update the instrumental record constrained ECS estimate from AR6. Since this was done many times before, we do not see a reason to justify it here.

- You said that you "*had to assume that variations in the pattern effect as estimated among the models and across the datasets are independent of each other*". It seems likely there is a correlation though, justifying your suggested next steps in the final sentence. You could do error propagation with assumed correlations between model and SST errors and a sentence or two could then say "strong correlation (r=whatever) does/doesn't greatly affect these conclusions...".

Basically your headline is "reduced uncertainty" and I'd like extra clarity on exactly what you're claiming.

Indeed, we assume that the uncertainties from model- and dataset- dependencies are independent. But the method applied also assumes that the ECHAM/MPI-ESM model is similarly different to other models in all other SST datasets as it is in the amip-piForcing simulation. To account for this we adjusted the mean values.

In the revised manuscript, we have updated the calculations based on the results from (Andrews et al., 2022). We now explicitly discuss this. We add a figure (Figure A4, shown below) in supplement which also shows the comparison between $\Delta\lambda$ derived from amip-piForcing simulations and difference in $\Delta\lambda$ between AMIPII and HadISST forced simulations from Andrews et al. (2022). As can be clearly seen there is no obvious correlation.

[Figure]

Figure 4

**Minor/typos**

Suggest extending subscripts to be absolutely clear, specifically "ECS_hist" (or "ECS_amip") to discriminate as you do with your $\lambda$ subscripts.

Thanks. We considered the suggestion. But since we clarify whenever stated in the manuscript that ECS is constrained based on the historical period we decided to retain "ECS".

P1L11 – 95 percentile -> 95th percentile

Done.

P1L19—20: "*To know what is required to meet the Paris Agreement goal it is imperative to better quantify and understand the rate of global warming*" You aren't really studying warming rates. Maybe something like "...it is imperative to better quantify and understand the ultimate amount of warming in response to a given forcing."

Replaced.

P2 lines messed up but between 25 and 30: "*It can be framed as N = F +λT, where N is the planetary energy imbalance which is generally measured as the net downward radiative flux at the top-of-the-atmosphere (TOA), F is the external radiative forcing, measured as the effective radiative forcing (ERF)*" I think you mean "defined" rather than "measured", both for TOA flux (even CERES is pinned to Argo etc for the mean) and ERF is clearly not measured either.

Thanks. In the revised manuscript, we have changed it to "defined".

P2 lines and equation: The equation uses $\Delta T$ and the text uses T both to denote changes in T. The same for N and F. I suggest using $\Delta T$, $\Delta F$ and $\Delta N$ consistently to denote changes.

T, N, F in equation represents the change in the parameters "relative to an unperturbed equilibrium climate state where $N = F = 0$". In the equation that gives ECS

$$\text{ECS} \approx \frac{F_{2\times}\Delta T}{\Delta F - \Delta N},$$

are the changes taken between two periods, in this case 1850-1900 and 2006-2019.

In the revised manuscript, to clarify we now add "...where the change in temperature, forcing and energy imbalance is taken between two periods, e.g. 1850-1900 and 2006-2019 denoted by $\Delta F$, $\Delta T$ and $\Delta N$."

P2L32—33:"To reconcile this discrepancy the community looked into the concept of pattern effect which is not taken into account in the traditional energy balance framework." This misses how changes in T data have reduced the apparent differences. Both Lewis Curry and Otto et al. used HadCRUT4. Pick some relevant citation – a recent e.g. is Clarke Richardson (2021, doi: 10.1029/2020EA001082) whose Table 2 shows how 1850—2019 ECS_hist estimates would increase relative to HadCRUT4. Cowtan Way infilling is worth about +10 %, while Berkeley Earth's more substantial improvements (more stations, infilling, improved sea ice treatment) give +17 % to ECS_hist. The authors could persuade me otherwise, but updated global temperature datasets seem both important context for a study based on inter-dataset issues.

Thanks for pointing to Clarke and Richardson (2021). We added "... In addition there has been upward revisions to the temperature record through in-filling (e.g. Clarke and Richardson,

2021), downward revisions to total forcing and small revisions to estimates of energy imbalance (Bellouin et al., 2020; Von Schuckmann et al., 2020; IPCC, 2021)."

P2L42: *"Atmosphere-Ocean General Circulation Model (AOGCM) simulations show that the long term climate response (response to abrupt4xCO2) resembles a temperature pattern similar to 'El- Nino Southern Oscillations' (ENSO)"* This could be interpreted to mean that models have shown/proven the "real" long-term pattern and should be rephrased to something like: AOGCMs' simulated pattern resembles ENSO. I think you wanted to make this point in P3L47 when you said "assumed correct", but I wasn't sure. Change from: *"...leads to more stabilizing feedback while an assumed correct opposite future temperature distribution would lead to a less stabilizing feedback"* To something like: *"...leads to more stabilizing feedback while the simulated future temperature distribution would lead to a less stabilizing feedback"*

Thanks. It is changed now. Also, we modify to "Atmosphere-Ocean General Circulation Model (AOGCM) simulates a long term climate response (response to $abrupt4xCO2$) that resembles a temperature pattern similar to 'El-Nino Southern Oscillations' (ENSO)"

Figure 1 caption: are your fixed-SST simulations really "AOGCM" which you've defined as "atmosphere ocean GCMs"? I interpreted AOGCM as including coupling to an ocean model.

*abrupt4xCO2, historical* are AOGCM while *observedSST-piForcing* are Atmosphere-only GCM simulations is written in the caption.

P3L57—58: "Such estimates are model dependent, nevertheless most models yield a dampening pattern effect based on the AMIPII dataset." Alone, "dampening effect" could simply mean negative feedback in general. I assume you mean dampening relative to the long-term abrupt4xCO2 pattern? If yes, please correct.

Thanks. We now write "Such estimates are model dependent, nevertheless most models yield a dampening pattern effect relative to the long-term $abrupt4xCO2$ pattern based on the AMIPII dataset (Andrews et al., 2018)."

Fig. 3 – second row first column figure is labelled "HadSST" rather than "HadISST"

Corrected.

P5L113—115: "For $\lambda_{hist}$, we take the changes in the global annual mean N and T from 1851-2014 of historical simulation relative to the mean of 1851-1900 of the same. Since, there are 10 ensemble members present in the historical simulation, we use the mean of the ensembles in our calculation" Can you rephrase? I think this is saying you took the mean of 1851-2014 minus the mean of 1841-1900 but am not certain. Also, did you mean "ensemble mean" instead of "mean of ensembles"?

In the revised manuscript, we have rephrased to "For $\lambda_{hist}$, we take the changes between the global 'annual' mean $N$ and $T$ of the entire period 1851-2014 of *historical* simulation relative to the mean of first 50 years of the same. Since, there are 10 ensemble members present in the *historical* simulation, we use the ensemble mean in our calculation.".

P8L142: is this the mean and standard error of the fits to each dataset, calculated using the standard Gaussian assumption, or something else?

The mean is the difference between $\lambda_{hist}$ and $\lambda$ from the *observedSST-piForcing* simulations. The standard error is calculated by the adding the errors from each in quadrature.

P8L145: rogue decimal point in error: 0.0.25

Done.

P8L151: typo "expectced" -> "expected"

Done.

P8L161: typo gap in 0 .59

Done.

P10Fig6: Are y axis units W m-2 K-1? Please add axis labels or explain why not.

Updated with y-axis units.

P11L195—196: "while the uncertainty is deduced from Figure 7.6 of this report." - "this report" could mean your paper, whereas I infer you mean "AR6 Figure 7.6"

Changed to "AR6 report".

P12L204: typo "bt" -> "by"

Done.

P14L245: Table A1 says that AMIPII = HadISST until Nov 1981, is that correct? If yes, then what are the slight differences in Fig. 1? If so, then is it surprising that there are such big, consistent differences in feedbacks in Fig. 7(a) when you've got multiple runs with only land/atmosphere variability?

Although before Nov 1981, AMIPII is same as HadISST1, re-basing the SST anomalies and subsequent adjustments changes the SST values prior to 1981. We have now modified with some details "Before Nov 1981, it uses HadISST1 while afterwards uses OI.v2 SST fields. The merging is done by first adding the HadISST1 anomalies relative to its own based period to OI.v2 climatology for the same base period and then adjusting that eliminates outliers and produce physically realistic SST values. The re-basing and adjustments changes relative temperature values prior to 1981 as well. "

**References**

Andrews, T., Gregory, J. M., Dong, Y., Armour, K., Paynter, D., Lin, P., Modak, A., Mauritsen, T., Cole, J., Medeiros, B., and et al. (2022). On the effect of historical sst patterns on radiative feedback. *Earth and Space Science Open Archive*, page 48.

Andrews, T., Gregory, J. M., Paynter, D., Silvers, L. G., Zhou, C., Mauritsen, T., Webb, M. J., Armour, K. C., Forster, P. M., and Titchner, H. (2018). Accounting for Changing Temperature Patterns Increases Historical Estimates of Climate Sensitivity. *Geophysical Research Letters*, 45:8490–8499.

Bellouin, N., Quaas, J., Gryspeerdt, E., Kinne, S., Stier, P., Watson-Parris, D., Boucher, O., Carslaw, K. S., Christensen, M., Daniau, A. L., Dufresne, J. L., Feingold, G., Fiedler, S., Forster, P., Gettelman, A., Haywood, J. M., Lohmann, U., Malavelle, F., Mauritsen, T.,

McCoy, D. T., Myhre, G., Mülmenstädt, J., Neubauer, D., Possner, A., Rugenstein, M., Sato, Y., Schulz, M., Schwartz, S. E., Sourdeval, O., Storelvmo, T., Toll, V., Winker, D., and Stevens, B. (2020). Bounding Global Aerosol Radiative Forcing of Climate Change. *Reviews of Geophysics*, 58(1):1–45.

Clarke, D. C. and Richardson, M. (2021). The Benefits of Continuous Local Regression for Quantifying Global Warming. *Earth and Space Science*, 8(5):1–21.

Gregory, J. M., Andrews, T., Ceppi, P., Mauritsen, T., and Webb, M. J. (2019). How accurately can the climate sensitivity to CO 2 be estimated from historical climate change? *Climate Dynamics*, 54(1-2):129–157.

IPCC (2021). *Climate Change 2021: The Physical Science Basis. Contribution of Working Group I to the Sixth Assessment Report of the Intergovernmental Panel on Climate Change*, volume In Press. Cambridge University Press, Cambridge, United Kingdom and New York, NY, USA.

Lewis, N. and Mauritsen, T. (2021). Negligible unforced historical pattern effect on climate feedback strength found in HadISST-based AMIP simulations. *Journal of Climate*, 34(1):39–55.

Olonscheck, D., Rugenstein, M., and Marotzke, J. (2020). Broad consistency between observed and simulated trends in sea surface temperature patterns. *Geophysical Research Letters*, 47(10):e2019GL086773.

Seager, R., Cane, M., Henderson, N., Lee, D.-E., Abernathey, R., and Zhang, H. (2019). Strengthening tropical Pacific zonal sea surface temperature gradient consistent with rising greenhouse gases. *Nature Climate Change*, 9(7):517–522.

Von Schuckmann, K., Cheng, L., Palmer, M. D., Hansen, J., Tassone, C., Aich, V., Adusumilli, S., Beltrami, H., Boyer, T., José Cuesta-Valero, F., Desbruyères, D., Domingues, C., Garciá-Garciá, A., Gentine, P., Gilson, J., Gorfer, M., Haimberger, L., Ishii, M., C. Johnson, G., Killick, R., A. King, B., Kirchengast, G., Kolodziejczyk, N., Lyman, J., Marzeion, B., Mayer, M., Monier, M., Paolo Monselesan, D., Purkey, S., Roemmich, D., Schweiger, A., Seneviratne, S. I., Shepherd, A., Slater, D. A., Steiner, A. K., Straneo, F., Timmermans, M. L., and Wijffels, S. E. (2020). Heat stored in the Earth system: Where does the energy go? *Earth System Science Data*, 12(3):2013–2041.

Watanabe, M., Dufresne, J. L., Kosaka, Y., Mauritsen, T., and Tatebe, H. (2020). Enhanced warming constrained by past trends in equatorial Pacific sea surface temperature gradient. *Nature Climate Change*, pages 3–9.

---

## Author Comment (AC2)

**Review of "Better constrained climate sensitivity when accounting for dataset dependency on pattern effect estimates" by Modak and Mauritsen.**

**Note- Our responses to the reviewer's comments are in red color text.**

Comment on egusphere-2022-976
Tim Andrews (Referee)

Referee comment on "Better constrained climate sensitivity when accounting for dataset dependency on pattern effect estimates" by Angshuman Modak and Thorsten Mauritsen, EGUsphere, https://doi.org/10.5194/egusphere-2022-976-RC2, 2022

This manuscript uses the MPI-ESM1.2-LR model to investigate the dependence of the pattern effect on underlying SST datasets used to force AGCMs with. By forcing the AGCM with 8 different reconstructions of historical SST patterns they find a substantial spread in the diagnosed pattern effect. By combining these results with published results on the inter-model spread the authors produce a new constraint on ECS from historical observations of past climate changes.

I found the manuscript well written and presented, and applaud the authors for tackling the important question of SST dataset dependency in methods that quantify the pattern effect. I think this manuscript will make a really useful contribution to the literature.

While there are a quite a comments here, all ought to be surmountable and are intended to be constructive.

We thank the reviewer for appreciating the study and for the constructive comments that helped improve the manuscript. The point by point responses are provided below.

**1 Major comments:**

1. Presentation of SST patterns: the authors present a useful analysis of the geographical patterns of SST trends across the different datasets, and show how they differ (e.g. Fig 3), but are unable to find a geographical region or principal explanation for the variation in feedback seen in their experiments (e.g. Fig 4 and Section 4.3). This is done by calculating the SST trend as a function of time (i.e. K /century). However I wonder if this is the most appropriate method to tackle the question being investigated. I think a more appropriate calculation (of more relevance to feedback and patterns effects) is the SST change per global-mean dT (analogous to how the feedback is calculated as radiative response per global-mean dT), rather than a trend in time. This normalised pattern has the additional advantage of removing any global-mean dT differences between the datasets (as shown in Figure 2).

Hence I recommend the authors repeat the analysis of Figure 3, Figure 5, Figure 6 and discussion in Section 4.3 but this time calculating the SST patterns per global-mean dT (ideally calculated like the feedback parameter, i.e. with linear regression, in this case regressing dT(lat,lon,t) against global-mean dT(t)). It might turn out to make little difference, in which that is fine – but it would be good to know and comment on if so. On the other hand, as it ought to relate better to the feedback and pattern effect it might improve the relationships in Fig 6. If so, I would recommend simply dropping the K/century trends and replacing all this with K(lat,lon)/ were <> denotes global-mean in this case.

Thanks. In the revised manuscript, we have updated the analysis of Figure 3, Figure 5 and Figure 6 by regressing dT(lat,lon,t) against global-mean dT(t) instead of temperature trends as a function of time. We find that the overall results remains similar.

2. Calculation of lambda4xCO2: the authors quote the 4xCO2 lambda for MPI-ESM1.2-LR as -1.48 +- 0.03 Wm-2 K-1 from regression in 150yrs of abrupt-4xCO2. However two other published papers (Andrews et al.,2022;https://doi.org/10.1029/2022JD036675) and Zelinka et al. (2020, GRL, updated to more models here https://zenodo.org/badge/latestdoi/259409949; see specifically https://github.com/mzelinka/cmip56_forcing_feedback_ecs/blob/master/CMIP6_ECS_ERF_fbks.txt) both report -1.39 Wm-2 K-1 for this model using similar linear regression on 150yrs of abrupt-4xCO2. This difference of 0.1 Wm-2 K-1 will feed into all estimates of the pattern effect, and is a significant % of the reported pattern effect, and could even change the sign in some cases. I wonder if the difference is because this manuscript calculates the 4xCO2 changes relative to the "mean of the last 500 years of piControl simulations", whereas both Andrews et al. and Zelinka et al. use the corresponding section of piControl and account for control drift? The authors ought to check, and I would recommend using the corresponding section and control drift method. If this is the cause of the difference, then this sensitivity to methodological choices ought to be discussed and acknowledged as a large source of uncertainty and potential bias in the pattern effect estimate.

We find that the difference is because of our use of surface temperature (ts) instead of surface air temperature as in Andrews et al. 2022 and Zelinka et al. 2020. IN the revised manuscript we write "Note, since we use surface temperature in our analysis, our estimate of $\lambda_{4xCO2}$ for MPI-ESM1.2-LR is larger than the estimate from Zelinka et al. (2020) and Andrews et al. (2022) which uses surface air temperature instead".

The difference in trends under global warming in surface and surface air temperature was investigated as part of IPCC AR6. Generally, models warm their surface air temperature by 5-10 percent more than the surface. However, observations of night time marine air temperatures co-located with surface temperatures show the opposite of similar magnitude. Therefore IPCC AR6 assessed the difference to be zero in all cases, however, with the issue that this causes some risk of confusion.

In this case we use the same definition throughout, so there is no comparing apples to bananas. There is one exception, though, when using Andrews et al. (2022) pattern effects, these are calculated based on surface air temperature. Therefore, the pattern effects and corresponding correction to ECS over that of AR6 is a bit higher by a few percent. The downward correction is then a bit smaller, in turn.

3. Discussion of the limitations/assumptions in the approach: when coming up with a combined estimate of the pattern effect, I think the manuscript might be making an assumption that isn't explicitly discussed. Specifically, if I've understood correctly, using the mean estimate and distribution from the observed-piForcing simulations is implicitly assuming that not only are all the various SST reconstructions independent (as the manuscript acknowledges, but clearly isn't true), but also that they are equally plausible reconstructions

of the truth. We do not know which dataset is 'best', right? The real world could have looked like any or none of them. If the authors have any thoughts on this, and how it could be taken forward – it might be useful to explicitly discuss. Related to this, I hesitate at statements such as line 223: " ... AMIP II dataset is an outlier suggesting that earlier studies may have overestimated the pattern effect...". If gives the impression (perhaps not intended) that the AMIP II results are not trustworthy, whereas we have no idea if AMIP II SSTs are any less likely than any other SST reconstruction. They might be fine. For example, just as a counter argument, there are other situations where simulations with AMIPII SSTs have been shown to perform better than simulations with HadISST. As Andrews et al. (2022) discuss: "Zhou et al. (2021;https://doi.org/10.1029/2022JD036675) showed that TOA radiative fluxes simulated by CAM5.3 correlated better with CERES observations when forced with AMIP II SSTs rather than HadISST SSTs, suggesting the results from amip-piForcing may be more reliable...". I do not think any major changes are required to address this comment, maybe just a simple change of wording or explicit acknowledgement of this point in the manuscript ought to suffice.

Thanks, yes we do not intend to discuss the validity of the reconstructions in this study, and so by outlier we mean in a statistical sense. In the revised manuscript, we now rewrite "...it turns out that the pattern effect estimated from the AMIPII dataset is by far the largest suggesting that earlier studies may have overestimated the pattern effect."

Also, we add "However, we reiterate that any of the SST dataset could be a possible path Earth could have taken."

4. Structural (model) dependence on SST dataset sensitivity: the authors acknowledge that their results may depend on the model used (MPI-ESM1.2), and I appreciate the effort in trying to combine the model spread and adjusting the mean from previous intercomparisons to account for this. However the authors should update the analysis to use numbers from larger model intercomparison of Andrews et al. (2022), rather than the Andrews et al. (2018) study. For amip-piForcing the pattern effect across 14 models is 0.70 +- 0.47 Wm-2 K-1 in Andrews et al. (2022), compared to 0.64 +- 0.40 Wm-2 K-1 in Andrews et al. (2018). So the difference in the mean to ECHAM6.3 ought to be slightly larger than the authors used here, and the spread slightly larger too. But moreover, the method applied assumes that the ECHAM/MPI-ESM model is similarly different to other model in ALL the datasets as it is in the amip-piForcing simulation. But we know from Andrews et al. (2022) that this is not so, the ECHAM and MPI-ESM pattern effects under hadSST-piForcing are much further from the mean than under amip-piForcing. For example, the pattern effect for ECHAM6.3 and MPI-ESM2.1-LL is 0.2 Wm-2 K-1 in hadSSTpiForcing, whereas the model mean is 0.48 Wm-2 K-1 (Andrews et al. 2022; Table 2). I'm not sure exactly what the solution is here, but it seems adjusting for the mean difference to ECHAM6.3 of just 0.1 Wm-2 K-1 based on Andrews et al. (2018) is insufficient. It ought to be larger based on the larger ensemble of Andrews et al. (2022) and larger again given the hadSST-piForcing results where ECHAM6.3 pattern effect is further away from the rest of the models. At a minimum this potential structural dependence of the results on the ECHAM6.3/MPI-ESM model ought to be explicitly discussed.

Thanks. In the revised manuscript, we have applied Andrews et al. 2022 and updated the calculations. We now write "We find the combined pattern effect estimate to be 0.37 $Wm^{-2}K^{-1}$ [-0.14 to 0.88 $Wm^{-2}K^{-1}$]. Our calculations for the combined estimate considers only the AMIPII based pattern effect estimates from Andrews et al. (2022). The combined estimate is obtained by subtracting the mean of ECHAM6.3 and MPI-ESM1.2-LR estimate in Andrews et al. (2022) from the sum of our estimate and their multi-model mean estimate. The uncertainty range in the combined estimate is deduced by adding in quadrature the standard deviations from our dataset dependent estimate and their multi-model estimate.

Thus, our calculation also accounts for the weaker pattern effect in ECHAM6.3 and MPI-ESM1.2-LR than in the multi-model mean. While accounting for the weaker pattern effect the assumption is that ECHAM6.3/MPI-ESM1.2LR is different from all the other models in all datasets as in AMIPII. However, one can argue this assumption. Inferred from Andrews et al. (2022), Figure A4 shows that not only ECHAM6.3/MPI-ESM1.2LR but also other models though produce larger pattern effects has a large difference in pattern effect estimates based on AMIPII and HadISST datasets."

It is certainly possible that MPI-ESM is more sensitive to changes in datasets than other models, however, this cannot be determined based on our study. We are therefore currently conducting a model intercomparison project to investigate this possibility further.

**2    Minor comments:**

- * Title: "Better constrained" does not particularly read well to me, how about "Improved constraints on.." ? But I'll let the authors decide.

  Thanks. We considered the suggestion but decided to stay with the original title.

- * Lines 45-49: I think it would be appropriate to include Andrews and Webb (2018; JCLIM,https://doi.org/10.1175/JCLI-D-17-0087.1) in the discussion of the mechanisms of the pattern effect.

  Thanks. Now it is included.

- * Lines 110: "-1.48 +- 0.03 Wm-2 K-1", what is the level error being presented here and throughout, 5-95% or something?

  We write in section *3. Model, datasets and experiments* "Unless otherwise specified, throughout the text, the displayed results are based on the mean of the 5 ensemble simulations while the uncertainty denotes the standard error from the ordinary least square (OLS) regression and the ranges are the 5-95$^{th}$ percentiles."
  Also, in section *4.5 An updated estimate of ECS* we add, "In this section, note that we consider the best estimate while the uncertainty (standard deviation) and ranges denotes the 5-95% confidence intervals."

- * Line 116: "... account for land surface warming in the fixed-SST simulation to calculate the forcing.." – the literature has various ways of doing this, it would be good to briefly clarify which approach was used.

  In the revised manuscript we clarify this and write "To calculate $F$, we account for the land surface warming: we subtract the product of $\lambda_{4xCO2}$ and the land surface warming from $N$ simulated by historical simulation in fixed-SST configuration (Hansen et al., 2005; Modak et al., 2018)."

- * Lines 126: "... pattern effect ranges from -0.01 +- 0.09 Wm-2 K-1..." – please clarify how the uncertainty is calculated here. Since it is difference between lambda_4xCO2 and lambda_piForcing have you added the errors in quadrature or something?

  Thanks. We now add "The uncertainty in the pattern effect estimates are calculated by adding the errors from $\lambda_{4xCO2}$ and respective $\lambda$ from *observedSST-piForcing* simulations in quadrature".

- * Line 129: "... the dataset mean pattern effect is lower than both Andrews et al. (2018) and the values considered in IPCC AR6..." – this reads as if the value is outside the range. So it is not quite right that the value is lower than those "considered" by Andrews et al. and IPCC since it is within their uncertainty range.

  We now write "The mean estimate of the pattern effect derived from different dataset is less than the multi-model mean estimate of Andrews et al. (2022) and the mean estimate considered in IPCC AR6 (IPCC, 2021)."

- * Line 200: ".. pattern effect estimate in MPI... averaged over all decades..." – I'm not exactly sure what this means, please clarify.

  We now write "When we apply our dataset dependent pattern effect estimate - mean of $\Delta\lambda$ deduced from each dataset, 0.20 $\text{Wm}^{-2}\text{K}^{-1}$ [0.01 to 0.39 $\text{Wm}^{-2}\text{K}^{-1}$]), the ECS is adjusted to 2.9 K [1.7 to 6.7 K] (Figure 8)."

- * Line 204: "bt" typo.
  Done.

- * Line 210-11: This sentence explains how the mean combined pattern effect estimate is arrived at, but it doesn't explain how the uncertainty is combined between the pattern effect dataset dependence in this manuscript and the multi-model spread, i.e. how is the -0.14 to 0.74 Wm-2 K-1 on line 210 arrived at?

  We now clarify and write "The uncertainty range in the combined estimate is deduced by adding in quadrature the standard deviations from our dataset dependent estimate and their multi-model estimate."

- * Appendix: I found the Table summarising the datasets and Figure A1 showing the 30yr moving window lambda useful, and would recommend integrating them into the main text. One option would be to replace Figure 7 in the main text with Fig A1, since I find this figure more useful and interesting. However I do appreciate this is somewhat personal preference, so I do not demand it and leave the authors to choose.

  To update ECS estimates based on the historical warming, the entire long-term period is generally applied, in this case 1871-2017. In Figure 7 we show how the long-term slope evolves when we increase the regression length by a year derived from different SST datasets. Hence, we find it relevant for this study and decide to retain Figure 7 in the main text. We agree that the sliding 30-year evolution of climate feedback is useful and interesting as it helps to identify how the 30-year feedback varies and can be used to compare the datasets. However, we do not use the 30-year periods to update ECS, and so is not central to our work. Hence, we decide to keep it in the supplementary.

I have signed the review.

Tim Andrews.

**References**

Andrews, T., Gregory, J. M., Dong, Y., Armour, K., Paynter, D., Lin, P., Modak, A., Mauritsen, T., Cole, J., Medeiros, B., and et al. (2022). On the effect of historical sst patterns on radiative feedback. *Earth and Space Science Open Archive*, page 48.

Hansen, J., Sato, M., Ruedy, R., Nazarenko, L., Lacis, A., Schmidt, G. A., Russell, G., Aleinov, I., Bauer, M., Bauer, S., Bell, N., Cairns, B., Canuto, V., Chandler, M., Cheng, Y., Del Genio, A., Faluvegi, G., Fleming, E., Friend, A., Hall, T., Jackman, C., Kelley, M., Kiang, N., Koch, D., Lean, J., Lerner, J., Lo, K., Menon, S., Miller, R., Minnis, P., Novakov, T., Oinas, V., Perlwitz, J., Perlwitz, J., Rind, D., Romanou, A., Shindell, D., Stone, P., Sun, S., Tausnev, N., Thresher, D., Wielicki, B., Wong, T., Yao, M., and Zhang, S. (2005). Efficacy of climate forcings. *Journal of Geophysical Research D: Atmospheres*, 110(18):1–45.

IPCC (2021). *Climate Change 2021: The Physical Science Basis. Contribution of Working Group I to the Sixth Assessment Report of the Intergovernmental Panel on Climate Change*, volume In Press. Cambridge University Press, Cambridge, United Kingdom and New York, NY, USA.

Modak, A., Bala, G., Caldeira, K., and Cao, L. (2018). Does shortwave absorption by methane influence its effectiveness? *Climate Dynamics*, 0(0):0.

Zelinka, M. D., Myers, T. A., McCoy, D. T., Po-Chedley, S., Caldwell, P. M., Ceppi, P., Klein, S. A., and Taylor, K. E. (2020). Causes of Higher Climate Sensitivity in CMIP6 Models. *Geophysical Research Letters*, 47(1):e2019GL085782.

---

## Author Response (AR2)

**Review of "Better constrained climate sensitivity when accounting for dataset dependency on pattern effect estimates" by Modak and Mauritsen.**

**Response to referee 1**

**Note- Our responses to the reviewer's comments are in red color text.**

The authors have addressed my concerns and I support publication. The new version includes the needed details and justifications for the selected methods, and the results progress our understanding of the pattern effect and its relationship to ECS.

Minor language comments are at the end.

SUMMARY OF RESPONSE In my first review, the comment on Seager et al. was phrased poorly, but the new Sec. 4.1 text fully addresses it.

Reviewer 2's suggestion of regressing local T versus global T is a better approach than the nonlinear suggestion I made. I'm glad the authors adopted it for Figs 3 and 5.

I want to thank the authors in particular for the new appendix figures. They are all useful for judging the analysis assumptions (A1,A4) or making it easy to compare with other research (A2).

Thanks. In the reviesed manuscript, we have made all the suggested changes.

GRAMMAR/TEXT CHANGES Sec. 5 could repeat that the post-1970s results are more negative in the mean, they don't just show smaller spread. Part of the community cares more about the modern period and a reminder could be helpful for them.

Thanks. The net feedback is more negative particularly for AMIPII after 1970s and not much with others. In the revised manuscript in section 5, we write "While the estimates from the 1970s until present are less dataset dependent, the major disparities originates in the early period and is driven by cloud feedback".

Minor grammar should be caught by a final check and copy editing but pure grammar corrections could sometimes change the meaning, such as by picking singular when I believe you intend plural. Please check: L48—50: "Whereas a colder tropical eastern Pacific and Southern Ocean, and a stronger tropical western Pacific warming is observed over the historical period, Atmosphere-Ocean General Circulation Model (AOGCM) simulates a long term climate response" –> "Atmosphere-Ocean General Circulation Models (AOGCMs) simulate. . ."?

Done.

L87: "The seven different observed-reconstructed SST datasets applied as boundary condition are HadISST1" –> "boundary conditions" plural?

Done.

L192 "To further investigate if any of the dataset bias the correlation," –> "any of the datasets" plural?

Done.

L202: "We find that the feedback from..." –> I think you mean plural "feedbacks" since Fig. 7 shows the different components?

Done.

Some bits are clunky and could perhaps be rephrased:

L106—108: "The temperature anomaly in ERSSTv5 forced observedSST-piForcing simulation is lower compared to others, consistent with Fueglistaler and Silvers (2021), is related to ship SST bias corrections made to the temperature during the 1880-1940s and 1950-1960s (Huang et al., 2017)" –> Consider splitting into two sentences or ",.. this is related to ship SST..."

Done.

L123—124: "we conduct a simulation in fixed-SST configuration, but evolving historical forcings to evaluate the effective radiative forcing (F) from 1851-2014." –> Consider after comma: "but with evolving historical forcings"

Done.

L178—179: "For instance, over the IPWP region, ERSSTv5 dataset shows this region is warming at the same rate as the globe but is true over the EEP region as well however with larger uncertainty." –> I think this sentence intends to say "this is also true over the EEP region"? Please check, and also convert to "the ERSSTv5 dataset" (added "the").

We thank the reviewer for asking us to check this figure. We find that there is an oversight. The figure in the manuscript shows the local warming to the warming over 50°S-50°N and not global warming which is what we wanted to show but wrongly stated in the caption and in the text. Now in the revised manuscript, we have corrected them. Now the caption replaces "global" to "50°S-50°N".
We now rewrite the text "We find that the local warming to the warming over 50°S-50°N from 1871-2017 in each of these regions have significant differences in some cases among the datasets (Figure 5). For instance, over the IPWP region, the ERSSTv5 dataset shows significant difference compared to the HadISST, COBE-SST2, had4krig, hadsst4krig or Vaccaro2021."

L231—233: ":Here, we apply their pattern effect based on AMIPII only as the number of models are larger which performed the amip-piForcing simulation." –> Suggest rephrasing the end bit to something like: "... based on AMIPII only as they had amip-piForcing simulations from more models."

Done.

**Review of "Better constrained climate sensitivity when accounting for dataset dependency on pattern effect estimates" by Modak and Mauritsen.**

**Response to referee 2**

**Note- Our responses to the reviewer's comments are in red color text.**

The authors have done a good job addressing reviewer comments. I have some final suggestions for the authors to consider before publication:

Thanks Tim for the appreciation.
The point by point responses are provided below.

1) Figure 5 and 6 and related discussion: I find it interesting that no strong correlation is found between the different SST pattern regions and the pattern effect. However I think the three most convincing SST metrics for explaining pattern effect variations haven't being analysed, and these are based on relative differences rather that single regions. For example Zhou et al. (2016) use the difference in warming between tropical ascending and descending regions, Andrews and Webb (2018) use the difference in south east-Pacific minus west-Pacific SST pattern and Fueglistaler and Silvers (2021) use SST# defined as the warming of the warmest 30% SSTs relative to the tropical mean. I do not demand the authors include these, but I do think it would be a useful addition (at least one of them) as it might be the reason for the poor correlation.

Indeed across all the datasets, the correlation between pattern effect and the regional warming relative to the the 50°S-50°N are of similar magnitude (0.64 over IPWP, 0.31 over EWP, -0.43 over EEP and 0.69 over SO). But as discussed in the manuscript the correlation is very strong over IPWP when ERSSTv5 dataset is not considered while calculating the correlation. In contrast, when Vaccaro21 is not considered, the correlation is only 0.15 over IPWP. In the manuscript, we discuss for other datasets as well "To further investigate if any of the datasets bias the correlation...".

Thanks. We apply the difference in south east-Pacific minus west-Pacific (SEP - WP) SST pattern following Andrews and Webb (2018) and calculated the correlation between the pattern effect and the SEP - WP $\Delta$T. We do not find a strong correlation across the datasets (Figure 1, shown below). We too think this is interesting and worth exploring in our planned multi-dataset and multi-model study.

In the revised manuscript we rewrite the final sentence of 2nd paragraph of section 4.3 "Thus, it is difficult to link the pattern effect variations across datasets only to the IPWP warming, rather we find all regions show a positive correlation with IPWP and SO showing a relatively stronger correlation than EWP and EEP (Figure 6)."

[Figure]

Figure 1

2) Line 154-155 and Fig A2: When comparing the total pattern effect from the CAM5.3 Green's function results (Lewis and Mauritsen, 2021) to the MPI-ESM simulations it is simply concluded that they are "substantially different", but I'm kind of left wondering "and so what"? What exactly is being said with this conclusion? At first I thought it implied an issue for the Green Functions method, but it's a different models Green's function and we know models simulate different pattern effects, so it is not entirely expected that the CAM5.3 green function results would give different estimates to the MPI simulations? So I guess I'm wondering what useful information is being drawn from this comparison? For example are the results substantially different in some interesting or unexpected way? Are the differences within the model-dependent uncertainty allowed for later on in the manuscript? If not, that is interesting and would suggest the uncertainty allowed for is not sufficient. I'm just throwing out ideas, but I think a couple more sentences are needed to explain how the authors interpret the "substantially different" result.

Thanks. In the revised manuscript we now add "We inferred the total pattern effect from Lewis and Mauritsen (2021) for the in-common SST datasets which they calculated based on CAM5.3 Green's function (Figure A2). We find that their estimates of total pattern effect are substantially different from our estimates for some of the SST datasets. The differences could be either because of the Green's function that is applied in Lewis and Mauritsen (2021) is derived from a different model (CAM5.3 Green's function applied to ECHAM6.3) and different models produce different pattern effects, or could be due to its inherent limitations (Zhou et al., 2017). However, we find that the uncertainty in the pattern effect estimates across the in-common SST datasets are of similar magnitudes between the studies. We plan to further address the comparison with the estimates derived from Green's function in a future study."

3) Figure 3: I got confused here. The pattern in dT(lat,lon) per dT ought to have a global-mean of unity, hence all the difference panels should have a global-mean of zero, right? But it doesn't

look that way. To eye some of them are strongly blue (negative). Is this just a visual thing and the global-mean really is zero? Or, is it a land effect, which is masked out here? Or something else?

In Figure 3, we are plotting the slope of the regression of temperature change evolution at each grid (lat,lon) against the global mean temperature change evolution. The global mean of the slope of regression need not be 1. So, the global mean of difference maps are non-zeros. We checked this. Thanks. If it was dt(lat,lon) divided by global mean dT, then global mean of this ratio has to be 1 and the mean of the difference maps has to be zero.

Minor comments:

4) Line 1 Abstract: I think this opening sentence requires a qualifier that it is referring to the best estimate ECS, since the range/uncertainty in ECS is still bigger than that deduced from other lines of evidence (e.g. see Sherwood et al., 2020).

Thanks. We now write "The best estimate of Equilibrium climate sensitivity (ECS) constrained based on the instrumental record of the..."

5) Line 245-247: "Figure A4 shows....." – I did not follow this sentence. Please try again.

In the revised manuscript, we rewrite "While accounting for the weaker pattern effect the assumption is that ECHAM6.3/MPI-ESM1.2LR is different from all the other models in all datasets as in AMIPII. However, one can argue this assumption. We infer from Andrews et al. (2022) and check this. We show in Figure A4 that not only ECHAM6.3/MPI-ESM1.2LR but also other models though produce stronger pattern effect has a similar difference in pattern effect estimates based on AMIPII and HadISST datasets as in ECHAM6.3/MPI-ESM1.2LR."

6) Line 280: append "... to address this outstanding concern"?

Appended.

I have signed the review. Tim Andrews.

**References**

Andrews, T., Gregory, J. M., Dong, Y., Armour, K., Paynter, D., Lin, P., Modak, A., Mauritsen, T., Cole, J., Medeiros, B., and et al. (2022). On the effect of historical sst patterns on radiative feedback. *Earth and Space Science Open Archive*, page 48.

Andrews, T. and Webb, M. J. (2018). The Dependence of Global Cloud and Lapse Rate Feedbacks on the Spatial Structure of Tropical Pacific Warming. *Journal of Climate*, 31(2):641–654.

Lewis, N. and Mauritsen, T. (2021). Negligible unforced historical pattern effect on climate feedback strength found in HadISST-based AMIP simulations. *Journal of Climate*, 34(1):39–55.

Zhou, C., Zelinka, M. D., and Klein, S. A. (2017). Analyzing the dependence of global cloud feedback on the spatial pattern of sea surface temperature change with a Green's function approach. *Journal of Advances in Modeling Earth Systems*, 9(5):2174–2189.